# Distinctly different active sites of ZnO-ZrO$_2$ catalysts in CO$_2$ and CO hydrogenation to methanol reactions

Jieqiong Ding [1,2,6], Yao Peng[3,6], Wei Xiong[2], Dongdong Wang[2], Ziran Xu[4], Qinxue Nie[2], Zheng Jiang [4], Zhi-Pan Liu [3,5], Cheng Shang [3,5] ✉ & Weixin Huang [2] ✉

The active site of a solid catalyst varies sensitively with the catalyzed reaction. Herein, using experimentally measured elementary surface reaction kinetics of CO$_2$ or CO hydrogenation reactions over a ZnO-ZrO$_2$ catalyst under working conditions in combinations with comprehensive structural characterizations and theoretical simulations, we unveil the distinctly different active sites in catalyzing the CO$_2$ or CO hydrogenation to methanol reaction. Zn$^{2+}$ cations with different local environments are present on the ZnO-ZrO$_2$ surface, including Zn$_1$ single atoms exclusively with a Zn-O-Zr local structure and Zn$_n$ clusters with both Zn-O-Zr and Zn-O-Zn local structures. The -Zr-O-Zr- structure bonded to the Zn$_n$ clusters is more easily to be reduced than that bonded to the Zn$_1$ single atoms. The Zn$_1$-single atom (-Zr-O-Zn-O-Zr-) is the active site for catalyzing the CO$_2$ hydrogenation to methanol reaction, whereas the Zn$_n$ cluster bonded to an in situ formed -Zr-V$_o$-Zr- structure (-Zn-O-Zn(-O-Zr-V$_o$-Zr-)-O-Zr-) is the active site for catalyzing the CO hydrogenation to methanol reaction. These results provide a reliable and effective methodology of elementary surface reaction kinetics for identifications of active sites of working catalysts in complex reactions and unveil how sensitively the active site structure varies with the catalyzed reaction.

Since the postulation of the concept of active site in 1925[1], delineating active sites in solid catalysts has become the central pursuit in the fundamental studies of heterogeneous catalysis[2] but meanwhile remained as an enduring challenge due to its inherent complexity[3-7]. The active site of a solid catalyst sensitively varies with the catalyzed reaction. Cu-ZnO based catalysts for CO$_x$ hydrogenation to methanol reactions are a well-known example. After long-

term studies and strong arguments, the in situ formed CuZn alloy[8-11] or Cu(I)$_{Cu}$-hydroxylated ZnO interface[12-19] were demonstrated by experimental evidence as the active structures of working Cu-ZnO based catalysts in the CO or CO$_2$ hydrogenation to methanol reactions, respectively. ZnO-ZrO$_2$ binary oxides have recently emerged as novel catalysts for the CO$_2$ hydrogenation to methanol reaction[20,21], and in combination with various types of zeolites, as highly active

[1]Hefei National Research Center for Physical Sciences at the Microscale, University of Science and Technology of China, Hefei, China. [2]State Key Laboratory of Precision and Intelligent Chemistry, iChEM, Key Laboratory of Surface and Interface Chemistry and Energy Catalysis of Anhui Higher Education Institutes, School of Chemistry and Materials Science, University of Science and Technology of China, Hefei, China. [3]Collaborative Innovation Center of Chemistry for Energy Material, Shanghai Key Laboratory of Molecular Catalysis and Innovative Materials, Key Laboratory of Computational Physical Science, Department of Chemistry, Fudan University, Shanghai, China. [4]National Synchrotron Radiation Laboratory, University of Science and Technology of China, Hefei, Anhui, China. [5]Shanghai Qi Zhi Institute, Shanghai, China. [6]These authors contributed equally: Jieqiong Ding, Yao Peng. ✉e-mail: cshang@fudan.edu.cn; huangwx@ustc.edu.cn

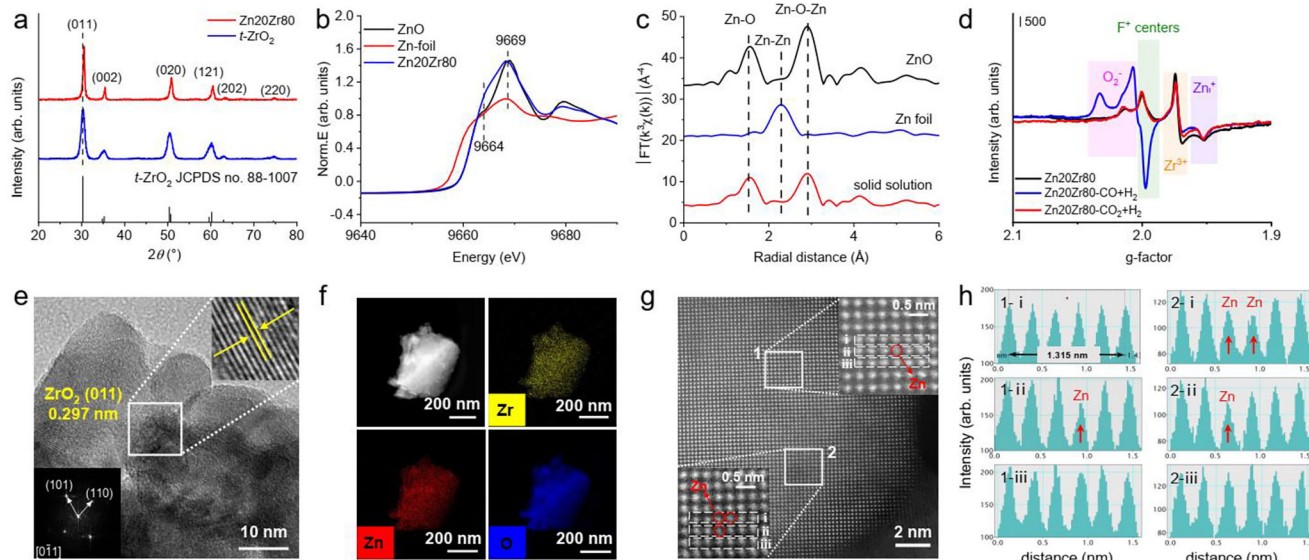

**Fig. 1 | Structural characterizations of Zn20Zr80 catalyst. a** X-Ray diffraction pattern of Zn20Zr80 and *t*-ZrO₂. **b** Zn K-edge XANES spectra and **c** Fourier transforms of the $k^3$-weighted Zn K-edge EXAFS spectra of Zn20Zr80 catalyst and referring ZnO and Zn foil. **d** ESR spectra of Zn20Zr80 calcined in Ar at 773 K, and subsequently treated under 3 MPa CO + H₂ (H₂:CO = 2) or 3 MPa CO₂ + H₂ (H₂: CO₂ = 3) at 573 K without exposure to air. All the spectra were measured at 140 K. **e** representative HRTEM image of Zn20Zr80 catalyst with an inserted electron diffraction pattern. **f** HAADF-STEM image and corresponding EDS mapping images of Zn20Zr80 catalyst. **g** Atomically resolved STEM images of edge areas of Zn20Zr80 nanoparticle, and **h** single-pixel line profiles across the atom rows of 1-i, 1-ii, 1-iii and 2-i, 2-ii, 2-iii within rectangles 1 and 2 marked in (**g**). The dark dots in 1 g and low-intensity single-pixel line profiles in 1 h correspond to Zn-containing atomic columns, marked with red circles and arrows, respectively.

bifunctional catalysts for the CO_x hydrogenation to value-added chemicals with methanol as the reaction intermediate[22–26], but the related active sites and reaction mechanisms are elusive. Different types of active structures were proposed in the CO₂[27–30] or CO[24,31] hydrogenation to methanol reactions. Meanwhile, although the formate (HCOO*) pathway is commonly accepted in the CO_x hydrogenation to methanol reaction[20,27,30], different types of formate species, bridging-adsorbed formate species (bri-HCOO*)[32] and tri-HCOO* species in a tetrahedral configuration[33], were argued as the active surface formate intermediate. Theoretical calculations also proposed a sequential hydrogenation mechanism of CO to methanol via HCO*, H₂CO*, H₃CO* intermediates[31]. Previous mechanistic studies of ZnO-ZrO₂ catalyzed CO_x hydrogenation reactions were mostly conducted at near atmospheric pressure[20,29,30,32,34] and occasionally at around 1.5 MPa[35,36], which are quite milder than the working conditions, whereas structures of oxide catalysts under the reductive CO_x hydrogenation reactions sensitively varied with the reaction pressure and temperature[37]. Thus, it is necessary to fundamentally investigate ZnO-ZrO₂ catalyzed CO_x hydrogenation reactions under the working conditions. Recently, elementary surface reaction kinetics of the formate hydrogenation reaction were successfully acquired on working oxide catalysts in the CO[38] or CO₂[39] hydrogenation to methanol reactions using temporal in situ DRIFTS spectroscopy, which, compared to the corresponding macroscopic reaction kinetics, unambiguously identified the active formate species.

In this work, via experimental studies of elementary surface reaction kinetics of CO_x hydrogenation reaction over a ZnO-ZrO₂ catalyst with an atomic Zn:Zr ratio of 1:4 (denoted as Zn20Zr80) under working conditions in combination with comprehensive structural characterizations and theoretical calculations, we unambiguously identify the Zn₁-single atom (-Zr-O-Zn-O-Zr-O-Zr-) and the partially-reduced Znₙ cluster with in situ formed -Zr-V_O-Zr- structure (-Zn-O-Zn(-O-Zr-V_O-Zr-)-O-Zr-O-Zr-) as the active sites for catalyzing the CO₂ and CO hydrogenation to methanol reactions, respectively. The active formate species and associate reaction mechanisms are also identified.

## Results

### Structural characterizations

A Zn20Zr80 catalyst synthesized via a co-precipitation method using Zr (NO₃)₄•5H₂O and Zn (NO₃)₂•6H₂O as the precursors[40] exhibits a Zn content of 20.3 at.% and a BET specific surface area of 23.1 m²/g. Its X-Ray diffraction (XRD) pattern (Fig. 1a) matches that of the standard pattern of pure tetragonal ZrO₂ phase (*t*-ZrO₂) (JCPDS file no. 88-1007), but the (011) spacing shifts to a higher angel than that of a reference pure *t*-ZrO₂ sample. This is an indication of substitutions of lattice Zr⁴⁺ cations by smaller-sized Zn²⁺ cations in Zn20Zr80. In the Zn K-edge X-ray absorption near-edge structure (XANES) spectra (Fig. 1b), Zn20Zr80 shows an almost identical peak at 9669 eV to pure ZnO but a stronger pre-edge shoulder peak at 9664 eV, which arise from to electron transitions from Zn 1s to Zn 4p-O 2p and Zn 4sp-O 2p hybridized states of the conduction band[41], respectively. The pre-edge feature of Zn K-edge XANES of ZnO was previously reported to increase as the ZnO particle size decreased[42,43]. In the corresponding Zn K-edge extended X-ray absorption fine structure Fourier transforms (FT-EXAFS) spectra (Fig. 1c, Supplementary Fig. 1, Supplementary Table 1), Zn20Zr80 shows the nearest neighboring Zn-O and Zn-Zn coordination shells of ZnO at approximately 1.8 Å (coordination number = 4) and 3.2 Å (coordination number = 12.2)[29], respectively. These Zn K-edge XAS results suggest the presence of ultrafine ZnO particles in Zn20Zr80 which are invisible in the XRD pattern. Thus, Zn20Zr80 consists of Zn²⁺-substituted *t*-ZrO₂ solid solution and fine ZnO particles, consistent with previous reports[29,34,35,44–46]. Electron paramagnetic resonance (ESR) spectra of Zn20Zr80 calcined in Ar at 773 K without exposure to air (Fig. 1d and Supplementary Fig. 2) only show signals of O₂⁻, F⁺ centers, Zr³⁺ and interstitial Znᵢ⁺ defects. The inverse EPR susceptibility $\chi_{EPR}$ of the observed ESR signals measured at different temperatures (Supplementary Fig. 3) were found to follow the Curie-Weiss law[47,48], confirming that these ESR signals arise from the pragmatically isolated sites. The observed ESR features barely change after an exposure of calcined Zn20Zr80 to air (Supplementary Fig. 4), suggesting that the F⁺ centers, Zr³⁺ and interstitial Znᵢ⁺ defects locate in the bulk. X-ray

Photoelectron Spectroscopy (XPS) spectrum (Supplementary Fig. 5) shows $Zn^{2+}$ and $Zr^{4+}$ cations on the Zn20Zr80 surface.

TEM images of Zn20Zr80 (Supplementary Fig. 6) show large aggregates consisting of fine nanoparticles, while careful analysis of HRTEM images (Fig. 1e and Supplementary Fig. 7) only gives the lattice fringes of $t$-$ZrO_2$ but not of ZnO, suggesting that the ultrafine ZnO particles probably embed in the bulk of Zn20Zr80 inaccessible by HRTEM characterizations. Low-resolution HAADF-STEM and corresponding elemental mapping images (Fig. 1f) demonstrate rather uniform distributions of Zr and Zn within Zn20Zr80, but examinations of the edge areas with atomically resolved STEM images and corresponding EDS mapping images (Fig. 1g and Supplementary Fig. 8) unveil an uneven Zn distribution on the Zn20Zr80 surface. The corresponding single-pixel line profiles across the atom rows in the inset of Fig. 1g (Fig. 1h) show columns with an average spacing of 0.260 nm corresponding to the {002} crystal planes of $t$-$ZrO_2$. The dominant lattice atoms are the bright Zr atoms, while a few lattice atoms are the dark Zn atoms. This directly visualizes the substitutions of lattice $Zr^{4+}$ cation by $Zn^{2+}$ cations. Moreover, such an analysis demonstrates the presence of $Zn_1$ single atom and $Zn_3$ cluster in the area 1 and 2, respectively. Similarly, the $Zn_2$ and $Zn_4$ clusters were also identified (Supplementary Fig. 9). Thus, the Zn20Zr80 surface is the $Zn^{2+}$-substituted $t$-$ZrO_2$ solid solution with the $Zn^{2+}$ cations of different local environments, including the $Zn_1$ single atom exclusively with the Zn-O-Zr local structure and the $Zn_n$ clusters with both the Zn-O-Zr and Zn-O-Zn local structures.

$CH_3OH$ adsorption was used to further probe the surface structure of Zn20Zr80 (Supplementary Fig. 10). Vibrational bands at 1156, 1060/1045, and 2871/1598/1374 $cm^{-1}$ appear upon $CH_3OH$ adsorption on Zn20Zr80 at RT, which, compared to those on the reference $t$-$ZrO_2$ and ZnO samples, can be assigned to the methoxy group at the $Zr^{4+}$ site ($CH_3O_{Zr}*$), methoxy group at the $Zn^{2+}$ sites with different local environments ($CH_3O_{Zn}*$), and bri-HCOO* species at the $Zn^{2+}$ sites, respectively. $CH_3OH$ dissociation is more extensive at the $Zn^{2+}$ site on the Zn20Zr80 surface than at the $Zr^{4+}$ site. Meanwhile, $CH_3OH$ dissociation is more extensive at the $Zn^{2+}$ site on the Zn20Zr80 surface that on the ZnO surface, but less extensive at the $Zr^{4+}$ site on the Zn20Zr80 surface that on the $t$-$ZrO_2$ surface. These also support that the Zn20Zr80 surface is the $Zn^{2+}$-substituted $t$-$ZrO_2$ solid solution.

## Catalytic performance in $CO_x$ hydrogenation reaction

Catalytic performance of Zn20Zr80 in $CO_2$ or CO hydrogenation reaction was evaluated. As the reaction temperature increasing from 523 to 573 K in the $CO_2$ hydrogenation reaction, the $CO_2$ conversion increases from 1.7% to 9.1% while the $CH_3OH$ selectivity decreases from 89.7 to 80.3% (Fig. 2a). The $CO_2$ conversion of 9.1% at 573 K is close to the equilibrium value (around 9.3%) under the adopted reaction condition[49]. We then measured the catalytic performance within the kinetics-controlled range (Supplementary Table 2). The derived Arrhenius plots (Fig. 2b and Supplementary Fig. 11) calculated from the $CH_3OH$ formation and $CO_2$ reaction rates give similar apparent activation energy($E_a$) of $81.9 \pm 1$ and $85.1 \pm 1$ kJ $mol^{-1}$, respectively,

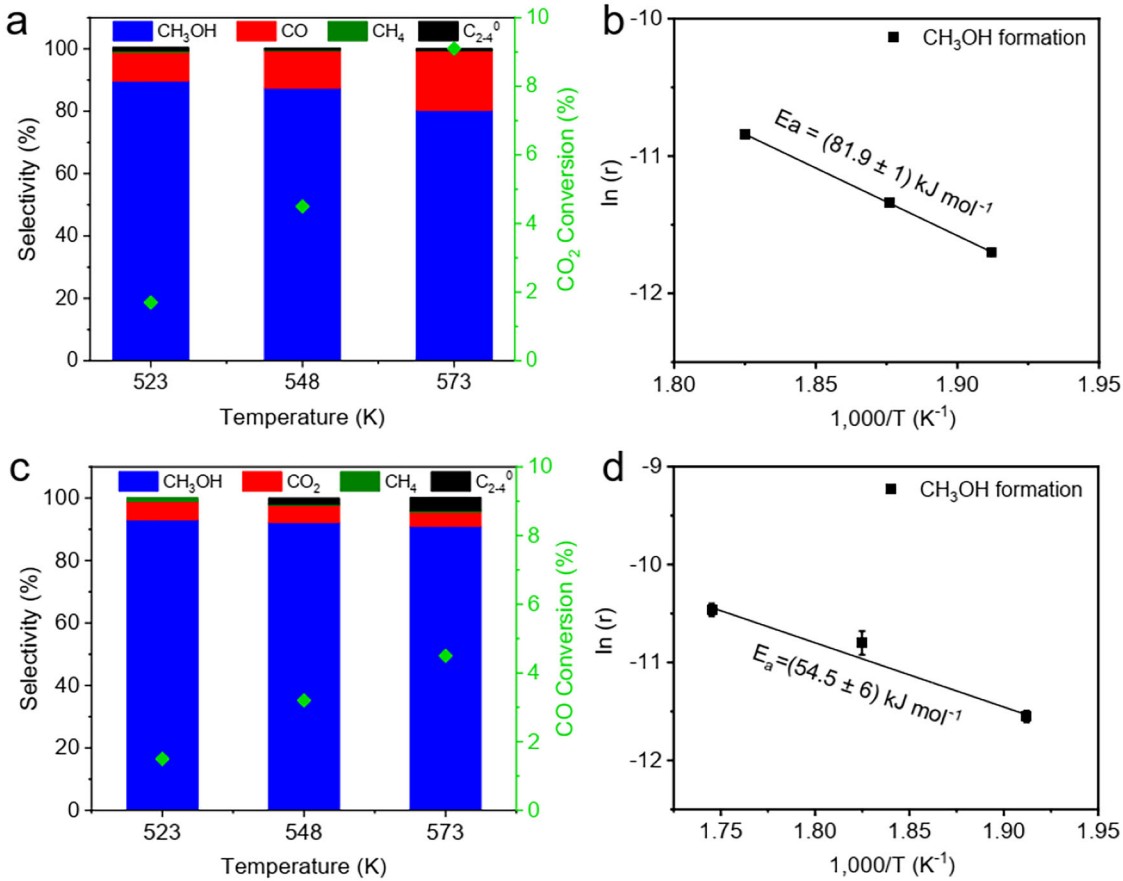

**Fig. 2 | Catalytic performance of Zn20Zr80 catalyst. a** Catalytic performance of Zn20Zr80 catalyst in $CO_2$ hydrogenation reaction (reaction condition: $H_2$: $CO_2$ = 3, 3 MPa; flow rate: 30 mL $min^{-1}$; catalyst mass: 1 g) and **b** Arrhenius plots of $CO_2$ hydrogenation to methanol reaction catalyzed by Zn20Zr80 catalyst derived from the data summarized in Supplementary Table 2 (reaction condition: $H_2$: $CO_2$ = 3, 3 MPa; flow rate: 30 mL $min^{-1}$; catalyst mass: 600 mg catalyst diluted with 400 mg SiC). **c** Catalytic performance of Zn20Zr80 catalyst in CO hydrogenation reaction (reaction condition: $H_2$: CO = 2, 3 MPa; flow rate: 30 mL $min^{-1}$; catalyst mass: 1 g) and **d** derived Arrhenius plots of CO hydrogenation to methanol reaction. The error bars in the figure represent the standard errors (SE) of the fitted values.

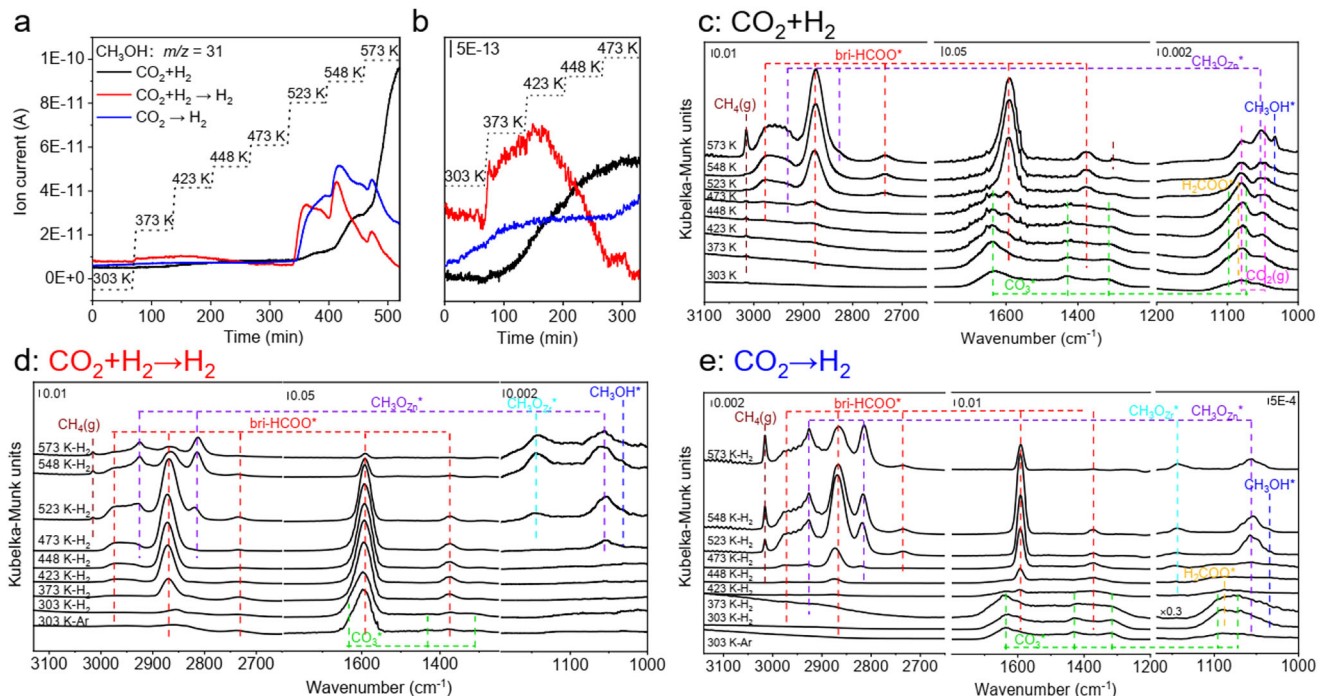

**Fig. 3 | Operando characterizations of CO₂ hydrogenation reaction. a** CH₃OH MS signals as a function of temperature over Zn20Zr80 under 3 MPa $CO_2$ + H₂ (H₂: CO₂ = 3) (CO₂ + H₂), Zn20Zr80 pretreated in 3 MPa $CO_2$ + H₂ at 573 K for 60 min, cooled to 303 K and purged in Ar under 3 MPa H₂ (CO₂ + H₂→H₂), and Zn20Zr80 pretreated in 3 MPa $CO_2$ at 303 K for 60 min and purged in Ar under 3 MPa H₂ (CO₂→H₂) and **b**, zoom-in curves between 303 and 473 K. Corresponding steady-state in situ DRIFTS spectra at indicated temperatures of **c**, CO₂ + H₂, **d**, CO₂ + H₂→H₂ and **e**, CO₂→H₂.

consistent with the literature results[50,51]. The apparent activation energy for CO production was also calculated as $125.9 \pm 7 \, kJ \, mol^{-1}$, much higher than that of CH₃OH formation. Thus, the mechanism of reverse water gas shift reaction followed by CO hydrogenation should barely contribute to the CH₃OH production by CO₂ hydrogenation reaction in our case, although it could not be fully excluded at high CO₂ conversions.

For the CO hydrogenation reaction, the CO conversion increases from 1.5 to 4.5% as the reaction temperature increasing from 523 to 573 K, while the CH₃OH selectivity decreases from 93.0% to 91.0% (Fig. 2c). The corresponding Arrhenius plots (Fig. 2d and supplementary Fig. 11) using CH₃OH formation and CO consumption rate give $E_a$ of $54.5 \pm 6$ and $54.3 \pm 6 \, kJ \, mol^{-1}$, respectively. The apparent activation energy for CO₂ production was also calculated as $37.3 \pm 8 \, kJ \, mol^{-1}$. Considering the low CO conversions and very low CO₂ selectivity in our case, the reaction pathway of CO₂ production and its consequent hydrogenation to methanol can be ignored in the CO hydrogenation to methanol reaction.

### Reaction mechanism of CO₂ hydrogenation reaction

ESR spectrum of Zn20Zr80 subjected to the CO₂ hydrogenation reaction at 573 K without exposure to air is almost identical to that calcined in Ar at 773 K (Fig. 1d), indicating that Zn20Zr80 is not reduced during the reaction. Additional oxygen vacancies were observed on ZnZrOₓ and spinel oxides under an H₂ atmosphere and can be quenched by a CO₂ environment[34,52], but few oxygen vacancies were generated on ZnZrOₓ in a CO₂ + H₂ atmosphere[36], consistent with our observations. Zn20Zr80-catalyzed CO₂ hydrogenation reaction was operando characterized via a combination of in situ DRIFTS and online mass spectroscopy mounted after the decompression valve of the in situ high temperature and high-pressure reactor cell for DRIFTS measurements. As shown in Fig. 3a, b, very weak gaseous CH₃OH signals was detected at a reaction temperature as low as 373 K, and the intensity increased with the reaction temperatures slowly up to 473 K

and then rapidly. Figure 3c and Supplementary Fig. 12 show the corresponding in situ DRIFTS spectra, and Supplementary Table 3 summarizes assignments of observed vibrational bands. Adsorbed CO₃* species dominates on the surface at 303 K. With the reaction temperature increasing, CO₃* slightly strengthens at 373 K, then keeps weakening, and disappears at 523 K; adsorbed H₂COO* species emerges at 373 K and keeps growing up to 448 K, then weakens and disappears at 523 K; adsorbed bri-HCOO* species emerges at 373 K and keeps growing up to 573 K; adsorbed CH₃O* species at the Zn site (CH₃O$_{Zn}$*) appears at 473 K and keeps growing up to 573 K; adsorbed methanol species (CH₃OH*) appears at 523 K and keeps growing up to 573 K.

A Zn20Zr80 catalyst pretreated in CO₂ hydrogenation at 573 K for 1 h, cooled to 303 K and purged in Ar exhibits the bri-HCOO*, CO₃* and CH₃OH* species whose hydrogenation reactivity was studied in 3 MPa H₂. Weak gaseous CH₃OH signals emerge as soon as the catalyst is heated, increases slowly but then decreases and disappears at 473 K, while strong CH₃OH productions occur again at 523 K and above (Fig. 3a, b). As shown in the corresponding in-situ DRIFTS spectra (Fig. 3d and Supplementary Fig. 13), CH₃OH* desorbs at low temperatures, giving the low-temperature CH₃OH productions; CO₃* disappears at 373 K while bri-HCOO* grows up to 473 K and then weakens up to 573 K; CH₃O$_{Zn}$* emerges at 473 K, grows up to 548 K and then weakens at 573 K; both CH₃O$_{Zr}$* and CH₃OH* emerge at 523 K, grow at 548 K and weakens at 573 K. These observations demonstrate the existence of a surface reaction pathway occurring above 473 K (high-barrier surface reaction pathway) for CO₂ hydrogenation to CH₃OH via surface intermediates of bri-HCOO*, CH₃O* and CH₃OH*, in which bri-HCOO* hydrogenation is the rate-limiting elementary surface reaction.

A Zn20Zr80 catalyst pretreated by 3 MPa CO₂ adsorption at 303 K and purged in Ar was found to exclusively exhibit CO₃* species whose hydrogenation reactivity was also studied in 3 MPa H₂. Weak gaseous CH₃OH signals emerge and then remain unchanged up to 473 K, then strong signals appear at 523 K and above (Fig. 3a, b). As demonstrated

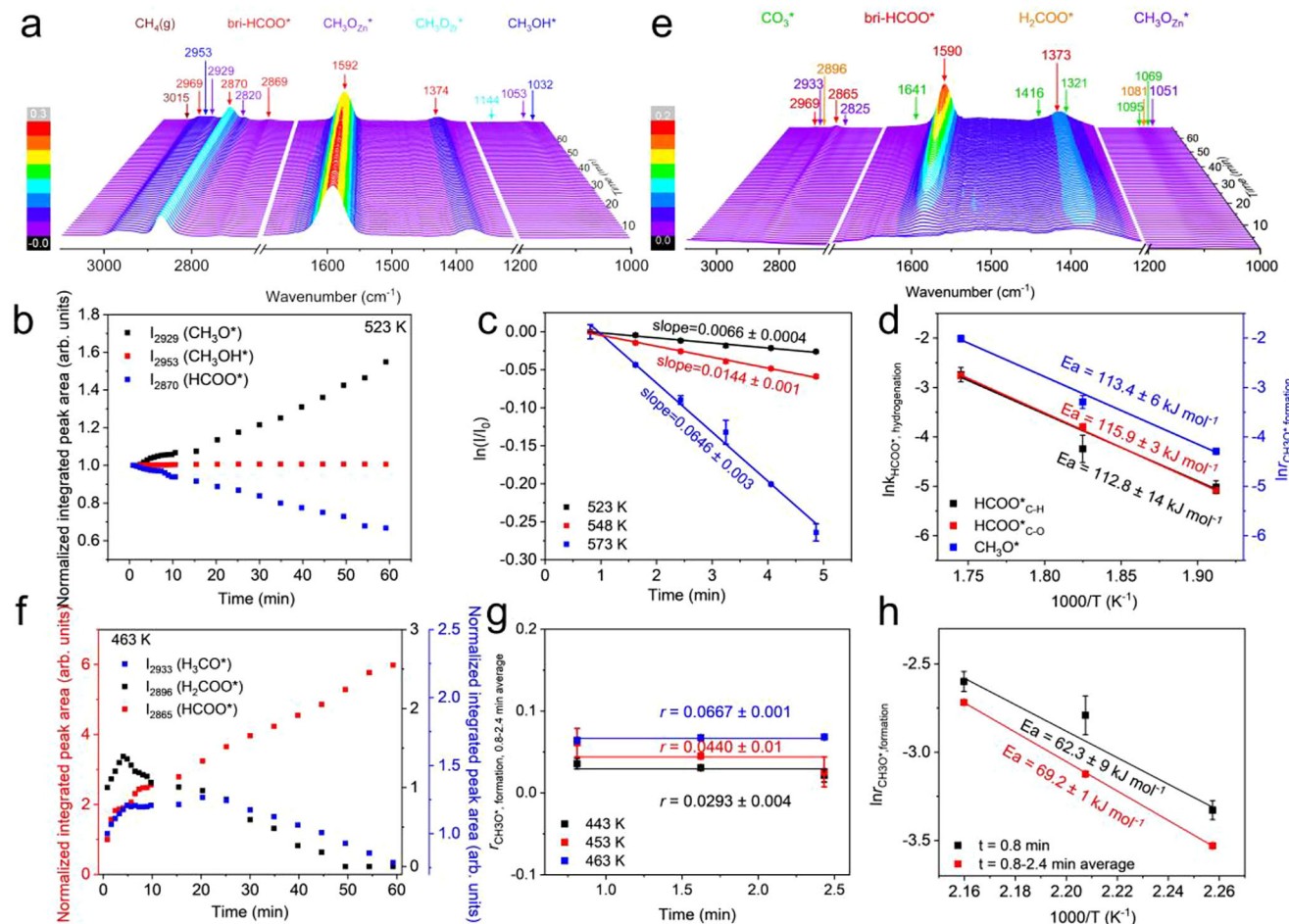

**Fig. 4 | Elementary surface reaction kinetics at high and low temperatures in CO₂ hydrogenation reaction. a** Temporal in situ DRIFTS spectra of Zn20Zr80 pretreated in 3 MPa CO₂ + H₂ at 573 K for 60 min and purged in Ar under 3 MPa H₂ at 523 K. **b** Corresponding normalized integrated peak areas of observed species in (**a**) as a function of time. **c** First-order reaction kinetic of bri-HCOO* hydrogenation reaction on Zn20Zr80 at different temperatures derived from panel (**b**), as well as Supplementary Figs. 24, 25. **d** Arrhenius plots of bridge formate hydrogenation and CH₃O* formation reactions derived from Fig. 4c and Supplementary Figs. 27–28,

respectively. **e** Temporal in situ DRIFTS spectra of Zn20Zr80 pretreated in 3 MPa CO₂ at 303 K for 60 min and purged in Ar under 3 MPa H₂ at 463 K. **f** Corresponding normalized integrated peak areas of various species in (**e**) as a function of time. **g**, the average rate of CH₃O* formation reaction on Zn20Zr80 at 0.8 to 2.4 min derived from panel (**f**). **h** Arrhenius plot of CH₃O* formation reaction using the CH₃O* formation rate at 0.8 min and the average rate of 0.8–2.4 min shown in Supplementary Fig. 38 and Fig. 4g. The error bars in the figure represent the standard errors (SE) of the fitted values.

in the corresponding in-situ DRIFTS spectra (Fig. 3e and Supplementary Fig. 14), CO₃* weakens greatly at 423 K and disappears at 473 K; bri-HCOO* emerges at 303 K, grow slowly up to 423 K and rapidly up to 548 K, then weakens at 573 K; H₂COO* obviously appears at 303 K, significantly grows at 373 K and then almost disappears at 423 K; both CH₃O_Zn* and CH₃OH* vary similarly to H₂COO* up to 423 K; additionally, CH₃O_Zn* re-emerges at 473 K, grows at 523 K and then weakens up to 573 K, while CH₃OH* re-emerges at 523 K, does not change much at 548 K and almost disappears at 573 K; CH₃O_Zr* emerges at 473 K and grows slowly up to 573 K. Thus, in addition to the high-barrier surface reaction pathway, another surface reaction pathway occurring at 303 K and above (low-barrier surface reaction pathway) also exists for CO₂ hydrogenation to CH₃OH via surface intermediates of CO₃*, bri-HCOO*, H₂COO*, CH₃O* and CH₃OH*, in which CH₃O* hydrogenation seems as the rate-limiting elementary surface reaction.

### Elementary surface reaction kinetics of CO₂ hydrogenation reaction

Using sodium formate as the formate source, a calibration of the DRIFTS signal from formate species was done according to the previously-reported method[53], whose results (Supplementary Fig. 15) confirm a linear correlation between the formate concentration and

the integrated area of IR vibrational peak. The elementary surface reaction kinetics were studied by measuring temporal evolutions of various surface intermediates during the hydrogenation reaction at different temperatures using in situ DRIFTS. For the high-barrier reaction pathway, the Zn20Zr80 catalyst pretreated in 3 MPa CO₂ + H₂ at 573 K for 60 min, cooled down to 303 K and then purged in Ar was rapidly heated in 3 MPa H₂ to desired temperatures. As shown in Fig. 4a and Supplementary Figs. 16, 17, bri-HCOO* is the only species observed on the catalyst surface and it barely changes at temperatures up to at 473 K but then keeps weakening at higher temperatures, leading to the emergence of CH₃O* and CH₃OH* species. The C-H stretch vibration region of temporal in situ DRIFTS spectra at various temperatures were peak-fitted to give the corresponding evolutions of bri-HCOO*, CH₃O* and CH₃OH* species (Supplementary Figs. 18–23), represented respectively by the vibrational features at 2870, 2929 and 2953 cm⁻¹, against the reaction time (Fig. 4b and Supplementary Figs. 24, 25). The decreasing of bri-HCOO* species, which arises from its hydrogenation reaction, accelerates with the reaction temperature increasing. CH₃O* keeps growing at 523 K, initially grows and then does not change much at 548 K, and initially grows and then weakens at 573 K. CH₃OH* remains unchanged at 523 K, initially remains unchanged and then weakens at 548 and 573 K. Following the elementary surface reaction

kinetic model proposed in Supplementary Fig. 26 with the bri-HCOO* hydrogenation reaction as the rate-limiting step, the initial growth of $CH_3O*$ species approximately equals to the $CH_3O*$ formation by the preceding bri-HCOO* hydrogenation reactions. The bri-HCOO* hydrogenation in 3 MPa $H_2$ were found to follow the first-order reaction kinetics (Fig. 4c), which is reasonable because the concentration of active H species, another reactant, could be considered constantly. The rate constants for bri-HCOO* hydrogenation reaction and the rates for $CH_3O*$ formation reaction at 523, 548 and 573 K (Fig. 4c and Supplementary Fig. 27) were derived and used to plot the Arrhenius plots (Fig. 4d), which give activation energies of bri-HCOO* hydrogenation reaction as $112.8 \pm 14$ kJ mol$^{-1}$ and of $CH_3O*$ formation reaction as $113.4 \pm 6$ kJ mol$^{-1}$. A similar analysis using the C-O vibrational feature of bri-HCOO* at 1590 cm$^{-1}$ (Supplementary Fig. 28) gives an activation energy of bri-HCOO* hydrogenation reaction as $115.9 \pm 3$ kJ mol$^{-1}$ (Fig. 4d). These results confirm that the bri-HCOO* species hydrogenates to produce the $CH_3O*$ species. However, the activation energy of bri-HCOO* hydrogenation reaction is much larger than the apparent activation energy of $CO_2$ hydrogenation to methanol reaction ($81.9 \pm 1$ kJ mol$^{-1}$), therefore, the high-barrier surface reaction pathway is not likely to be responsible for the methanol synthesis from $CO_2$ hydrogenation catalyzed by Zn20Zr80 catalyst.

For the low-barrier reaction pathway, the Zn20Zr80 catalyst pretreated in 3 MPa $CO_2$ at 303 K for 60 min and then purged in Ar was rapidly heated in 3 MPa $H_2$ to 443, 453 and 463 K at which the high-temperature surface reaction pathway barely works. As shown in Fig. 4e and Supplementary Fig. 29, the temporal evolutions of $CO_3*$, bri-HCOO*, $H_2COO*$ and $CH_3O*$ species depend on the reaction temperatures. The C-H stretch vibration region of temporal in situ DRIFTS spectra at various temperatures were peak-fitted (Supplementary Figs. 30–35) to give the corresponding evolutions of bri-HCOO*, $H_2COO*$ and $CH_3O*$ species, represented respectively by the vibrational features at 2865, 2896 and 2933 cm$^{-1}$, against the reaction time (Fig. 4f and Supplementary Figs. 36, 37). bri-HCOO* keeps increasing at all temperatures, whereas $H_2COO*$ initially increases but then decreases. These observations clearly demonstrate the existence of another type of bri-HCOO* species (denoted as bri-HCOO*-I) which exhibits vibrational features indistinguishable with those of the bri-HCOO* species observed in the high-barrier reaction pathway (denoted as bri-HCOO*-II) in our DRIFTS spectra but is capable of hydrogenating at low temperatures. $CH_3O*$ initially grows, then does not change at 443 and 453 K but weakened at 463 K. The $CH_3OH*$ species was not observed, probably due to the very low coverage. Following elementary surface reaction kinetic model proposed in Supplementary Fig. 39 with the $CH_3O*$ hydrogenation as the rate-limiting elementary surface reaction, the initial growth of $CH_3O*$ species approximately equals to the $CH_3O*$ formation by the preceding bri-HCOO*-I hydrogenation reactions. The rates for the $CH_3O*$ formation reaction at 443, 453 and 463 K were derived (Fig. 4g) and used to plot the Arrhenius plot (Fig. 4h) respectively, which gives activation energies varying between $62.3 \pm 9$ and $69.2 \pm 1$ kJ mol$^{-1}$, slightly lower than the apparent activation energy of $CO_2$ hydrogenation to methanol ($81.9 \pm 1$ kJ mol$^{-1}$). Thus, the low-barrier surface reaction pathway is predominantly responsible for the methanol synthesis from $CO_2$ hydrogenation catalyzed by the Zn20Zr80 catalyst.

It should be noted that the surface reactions of various surface intermediates on working Zn20Zr80 involved in the low-barrier reaction pathway proceed fast at typical reaction temperatures above 448 K, leading to their low coverages which are beyond the detection sensitivity of in situ DRIFTS during our studies. Nevertheless, the catalytic performance of Zn20Zr80 is predominantly contributed by the low-barrier reaction pathway. The surface intermediates on working Zn20Zr80 above 448 K observed by in situ DRIFTS are those involved in the high-barrier reaction pathway, which, however, are spectators.

## Theoretical simulations of $CO_2$ hydrogenation reaction

Comprehensive theoretical calculations were then carried out to examine the thermodynamics of the Zn-ZrO$_2$ surface configurations and possible $CO_2$ hydrogenation reaction pathways utilizing our recently-developed large-scale machine learning atomic simulation[54] and DFT computation. In accordance with the experimental observations of $Zn_1$ single atoms and $Zn_n$ clusters on Zn20Zr80 oxide solid solution catalyst with a t-$ZrO_2$ phase, we adopted a slab model of the t-$ZrO_2$ (101) surface and replaced 1 to 3 Zr atoms on the surface with Zn, thus creating ZnZrO surfaces with Zn coverage percentages of 8.3%, 16.6%, and 25%, denoted as $Zn_1$-$ZrO_2$, $Zn_2$-$ZrO_2$, and $Zn_3$-$ZrO_2$, respectively. The surface O atoms were removed accordingly to maintain Zn with a valence of +2 and Zr with a valence of +4.

By exploring over 10,000 minima for each composition using SSW-NN, we obtained the global minimums of $Zn_1$-$ZrO_2$, $Zn_2$-$ZrO_2$ and $Zn_3$-$ZrO_2$ surfaces as shown in Fig. 5a–c, respectively. $Zn_3$-$ZrO_2$ was calculated more stable than $Zn_1$-$ZrO_2$ by 20.2 kJ mol$^{-1}$ per Zn atom (Eq. (18)), suggesting a tendency for ZnO species to aggregate on $ZrO_2$ surfaces, in alignment with earlier simulation findings[31]. A more detailed examination reveals the presence of two distinct Zn sites: Zn-O-Zr ($Zn_{Zr}$) in all three $Zn_n$-$ZrO_2$ configurations and Zn-O-Zn ($Zn_{Zn}$) exclusively in $Zn_3$-$ZrO_2$. On the $Zn_1$-$ZrO_2$ surface (Fig. 5a), the most favorable location for introduced oxygen vacancies is in close proximity to the monodispersed Zn site. The coordination number of the neighboring surface Zn and two Zr atoms are 4 and 6, respectively, while the coordination number of other surface Zr atoms remains as 7, consistent with the pristine t-$ZrO_2$ (101) surface. On the $Zn_2$-$ZrO_2$ surface (Fig. 5b), the Zn atoms remain monodispersed, forming a linear arrangement along the [010] direction. Due to the increased density of monodispersed Zn atoms, $Zn_{4c}$ and $Zr_{6c}$ are neighbors. On the $Zn_3$-$ZrO_2$ surface (Fig. 5c), the $Zn_{Zn}$ site could be regarded as aggregated ZnO species, with a Zn-Zn distance of 3.39 Å, slightly longer than that in bulk ZnO (3.21 Å). The aggregation of ZnO on the surface results in additional oxygen vacancies, leading to all surface Zr atoms adopting a $Zr_{6c}$ coordination.

The potential reaction pathways of $CO_2$ hydrogenation to methanol under 3 MPa $CO_2$ + $H_2$ ($H_2$: $CO_2$ = 3) at 573 K were explored at both the $Zn_{Zr}$ and $Zn_{Zn}$ sites through comprehensive DFT calculations of the Gibbs free energy (Eq. (17)). $H_2$ preferentially dissociates heterolytically at the $Zn_{Zr}$-O pair of the Zn-$ZrO_2$ surfaces to form H*-$Zn_{Zr}$ and H*-O species with a low barrier of 23.2 kJ mol$^{-1}$. At the $Zn_{Zr}$ site on the $Zn_1$-$ZrO_2$ surface (Fig. 5d and Supplementary Table 5), the reaction commences with $CO_2$ adsorption at the $Zr_{6c}$ site and $H_2$ dissociation at the $Zn_{Zr}$-O pair. Subsequently, the H*-$Zn_{Zr}$ species reacts with $CO_2*$ by surmounting a barrier of 34.7 kJ mol$^{-1}$ (Fig. 5d, TS1), forming a bri-HCOO$_{Zr,Zr}$* intermediate with the O atom bonded to two $Zr_{6c}$ cations. The bri-HCOO$_{Zr,Zr}$* species further hydrogenates with another H*-$Zn_{Zr}$ species with a barrier of 30.9 kJ mol$^{-1}$ to form the $H_2COO*$ intermediate (TS2). Then the $H_2COO*$ species hydrogenates with a weakly-adsorbed $H_2$ species into H*-$Zn_{4c}$ and $H_2COOH*$ with a low barrier of 11.6 kJ mol$^{-1}$ (TS3). The $H_2COOH*$ then reacted with H*-$Zn_{Zr}$, forming OH* and $CH_3O*$ with a barrier of 77.2 kJ mol$^{-1}$ (TS4), which are then hydrogenated by overcoming a barrier of 41.5 kJ mol$^{-1}$ (TS5) to $H_2O$ and 86.8 kJ mol$^{-1}$ (TS6) to methanol as the final product, respectively. On the $Zn_2$-$ZrO_2$ surface of a similar local Zn environment to the $Zn_1$-$ZrO_2$ surface, $H_2$ dissociation preferentially occurs at the $Zn_{Zr}$-O site, and interestingly, endows the identical $Zr_{6c}$ sites A and B on a pristine surface (Fig. 5b) different local environment. Site A emerges on the opposite side of the oxygen vacancy, counter to the H*-$Zn_{Zr}$ species, whereas the site B positions adjacently to the neighboring H*-$Zn_{Zr}$ species. This leads to $CO_2$ adsorption with the presence of H*-$Zn_{Zr}$

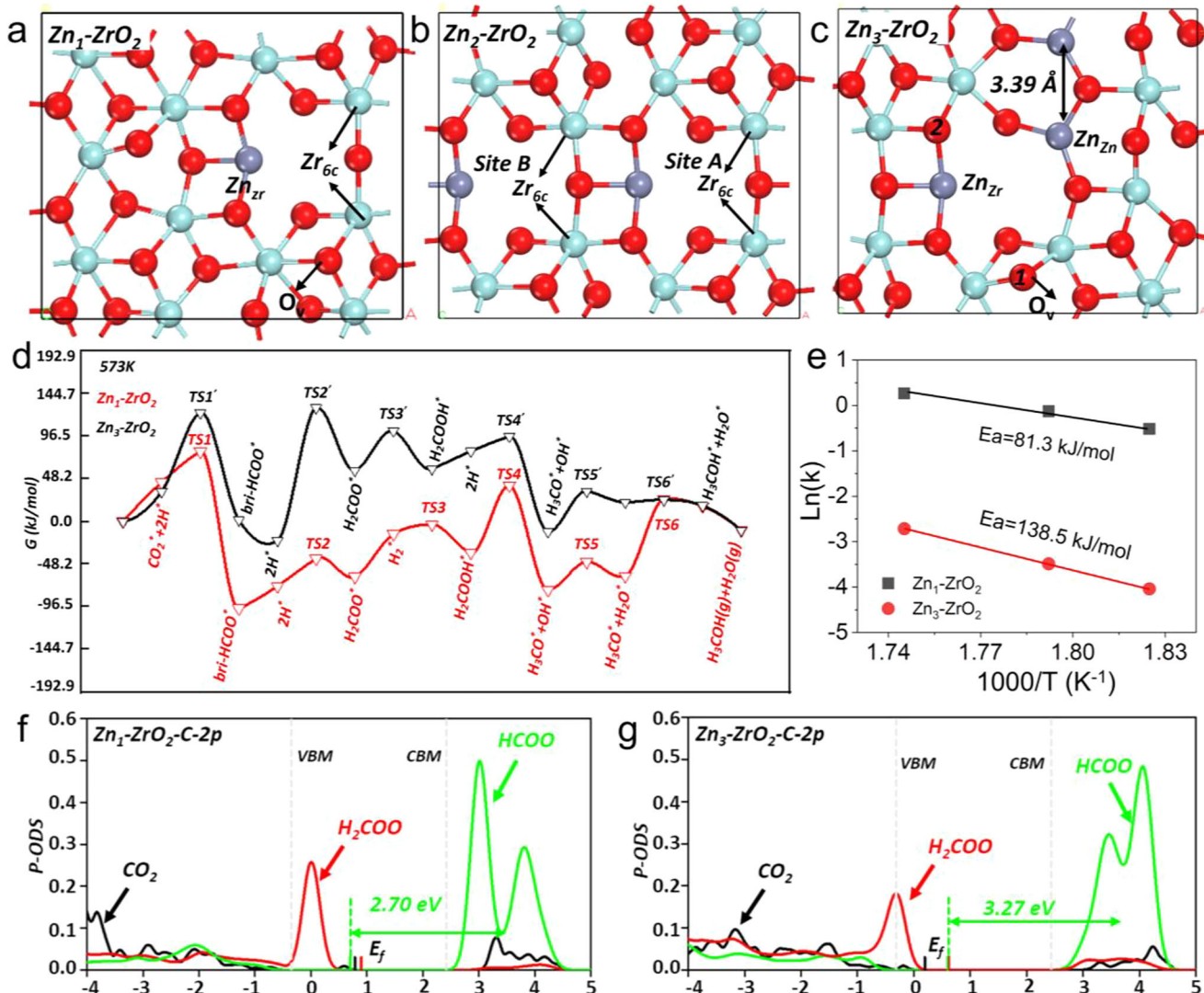

**Fig. 5 | Theoretical calculations of CO₂ hydrogenation reaction. a–c** Local snapshot of surface configurations of three global minimum structure with different surface Zn concentration, i.e., $Zn_1$-$ZrO_2$, $Zn_2$-$ZrO_2$ and $Zn_3$-$ZrO_2$. The $O_v$ is the most stable site when the lattice O is removed. Key reaction sites are labeled out in each (**d**), The Gibbs free energy profile of $CO_2$ hydrogenation on two catalysts under reaction condition, i.e., $Zn_1$-$ZrO_2$ (red curve, 573 K, 3 MPa, $H_2$: $CO_2$ = 3) and $Zn_3$-$ZrO_2$ (black curve, 573 K, 3 MPa, $H_2$: $CO_2$ = 3). **e** Logarithm of methanol production rates versus the reciprocal of temperatures on two catalysts. **f, g** The $p$DOS for the $2p$ orbitals associated with each carbon species ($CO_2^*$, $HCOO^*$, and $H_2COO^*$) on both $Zn_1$-$ZrO_2$ and $Zn_3$-$ZrO_2$ surfaces. All the DOSs are aligned via the Zr $3d$ states of the bulk $t$-$ZrO_2$. The $E_f$ indicate the Fermi level; the vertical gray dotted lines indicate the valence band minimum (VBM) and conduction band minimum (CBM) of bulk $ZrO_2$, respectively.

species at site A with an adsorption energy of 107.1 kJ mol⁻¹ while at site B with an adsorption energy of 137 kJ mol⁻¹. The reaction between H*-$Zn_{Zr}$ and $CO_2^*$ at site A or site B into the bri-$HCOO_{Zr,Zr}^*$ exhibits an energy barrier of 85.9 and 139.9 kJ mol⁻¹, respectively. The energy profiles for the further hydrogenation reactions of bri-$HCOO_{Zr,Zr}^*$ species at site A into $CH_3O^*$ are similar to on the $Zn_1$-$ZrO_2$ surface. On the $Zn_3$-$ZrO_2$ surface (Fig. 5d and Supplementary Table 6), $H_2$ dissociation also occurs at the $Zn_{Zr}$-O site, and $CO_2$ adsorbs bridgingly at the $Zn_{Zr}$-O-$Zr_{6c}$ site. Then the $CO_2^*$ species undergoes stepwise hydrogenation reactions into the bri-$HCOO_{Zn,Zr}^*$, $H_2COO^*$, $H_2COOH^*$, and $H_3CO^*$ intermediates, in which the rate determining step and the overall barrier is the $HCOO_{Zn,Zr}^*$ hydrogenation by H*-$Zn_{Zr}$ with a barrier as high as 146.7 kJ mol⁻¹.

Thus, although $CO_2$ hydrogenation to methanol follows the same reaction pathway on $Zn_1$-$ZrO_2$, $Zn_2$-$ZrO_2$ and $Zn_3$-$ZrO_2$, the rate-limiting step and associated barrier differ much. It is the hydrogenation of $H_2COOH^*$ into $CH_3O^*$ and $OH^*$ with a barrier of 77.2 kJ mol⁻¹ and the hydrogenation of $CH_3O^*$ into $CH_3OH^*$ with a

barrier of 86.8 kJ mol⁻¹ on $Zn_1$-$ZrO_2$, the hydrogenation of $CO_2^*$ into bri-$HCOO_{Zr,Zr}^*$ with a barrier of 85.8 kJ mol⁻¹ and the hydrogenation of $CH_3O^*$ into $CH_3OH^*$ with a barrier of 96.5 kJ mol⁻¹ on $Zn_2$-$ZrO_2$, and the hydrogenation of bri-$HCOO_{Zn,Zr}^*$ into $H_2COOH^*$ with a barrier of 146.7 kJ mol⁻¹ on $Zn_3$-$ZrO_2$. Microkinetic simulations for $CO_2$ hydrogenation to methanol on $Zn_1$-$ZrO_2$ and $Zn_3$-$ZrO_2$ catalysts using the free energetics derived from DFT give an apparent activation energy of 81.3 kJ mol⁻¹ on $Zn_1$-$ZrO_2$ and of 138.5 kJ mol⁻¹ on $Zn_3$-$ZrO_2$ (Fig. 5e). These results suggest that the experimentally-observed low-barrier surface reaction pathway of $CO_2$ hydrogenation to methanol, in which the $CH_3O^*$ formation reaction from the bri-$HCOO^*$-I species exhibits an activation energy of 69.6 kJ mol⁻¹ and the $CH_3O^*$ hydrogenation reaction is the rate-limiting step, probably occurs at the $Zn_1$ single atom site of Zn20Zr80, while the experimentally-observed high-barrier surface reaction pathway, in which the bri-$HCOO^*$-II hydrogenation is the rate-limiting step with an activation energy of 112.8 kJ mol⁻¹, probably occurs at the $Zn_n$ cluster sites. The experimentally-observed bri-$HCOO^*$ species during $CO_2$

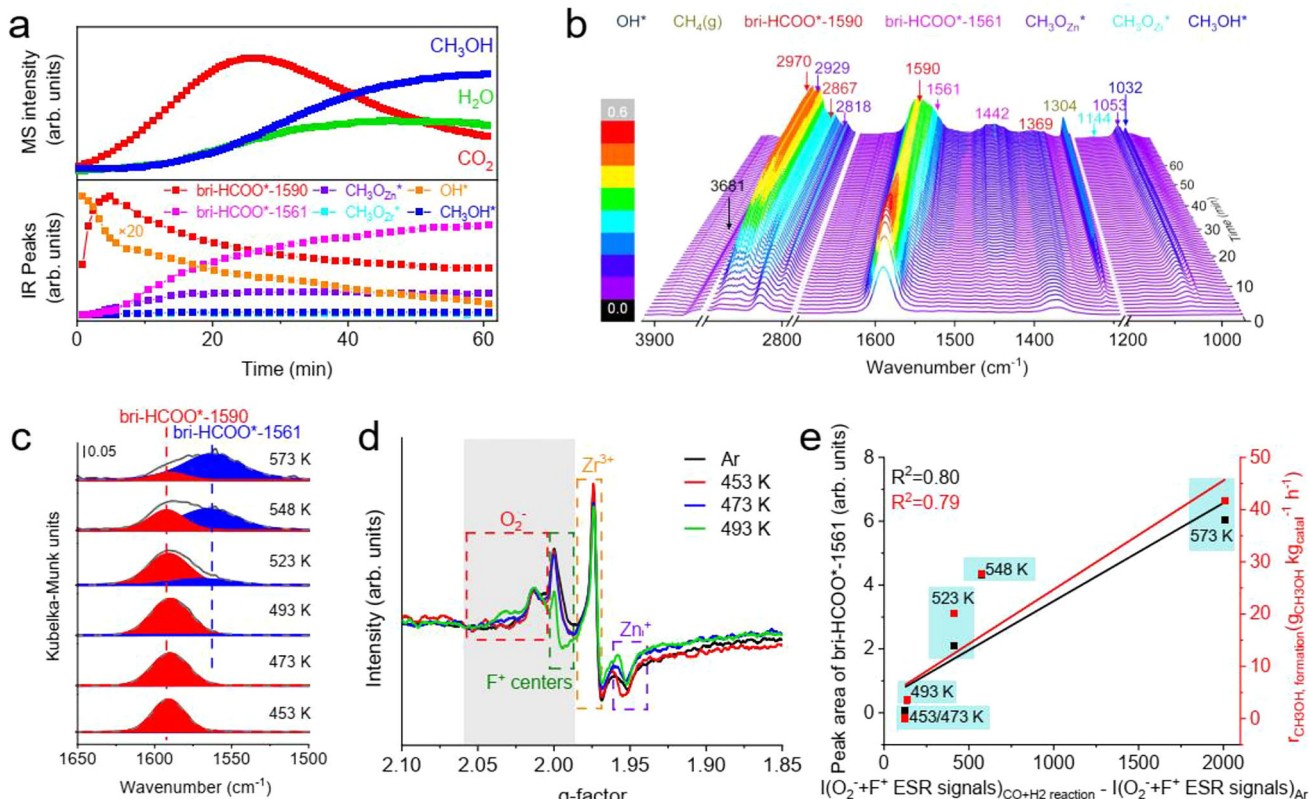

**Fig. 6 | Operando characterizations and in situ restructuring in CO hydrogenation reaction. a** CH₃OH MS signals and corresponding IR peaks of species in (**b**) as a function of time, and **b**, corresponding temporal in situ DRIFTS spectra of Zn20Zr80 under 3 MPa CO + H₂ (H₂:CO = 2) atmosphere at 573 K. **c** Peak-fitted C-O vibrational region of steady-state in situ DRIFTS spectra of Zn20Zr80 under 3 MPa CO + H₂ at indicated temperatures. **d** ESR spectra measured at 140 K of Zn20Zr80

calcined in Ar at 773 K and subsequently treated under 3 MPa CO + H₂ at indicated temperatures without exposure to air. **e**, intensity of vibrational feature of bri-HCOO*-1561 species on Zn20Zr80 treated under 3 MPa CO + H₂ at different temperatures and CH₃OH formate rate of Zn20Zr80-catalyzed CO hydrogenation reaction at different temperatures as a function of corresponding amount of created O₂⁻ and F⁺ centers on Zn20Zr80 surface.

hydrogenation on Zn20Zr80 contains the reactive bri-HCOO$_{Zr,Zr}$* (bri-HCOO*-I) species below 448 K while is dominantly the much less reactive bri-HCOO$_{Zn,Zr}$* (bri-HCOO*-II) species above 448 K. Additional theoretical calculations using PBE-D3[55] to include the dispersion interaction (Supplementary Table 7) show that the dispersion interaction does not affect the computed reaction energies much.

To gain insights into the disparate catalytic behaviors of Zn₁-ZrO₂ and Zn₃-ZrO₂ surfaces, we performed the electronic structure analysis of various adsorbed species. Upon H₂ dissociation on both surfaces, the H* of H*-Zn$_{zr}$ acquires a negative charge of around -0.4|e|, with the bonded Zn also gaining -0.2|e|, while the H* of H*-O acquires a positive charge of about 0.6|e|. Thus, the reactivity of H* species was similar on Zn₁-ZrO₂ and Zn₃-ZrO₂. The Projected Density of States (pDOS) for the 2p orbitals associated with the CO₂*, HCOO*, and H₂COO* on Zn₁-ZrO₂ and Zn₃-ZrO₂ were analyzed (Fig. 5f, g). Notably, an unoccupied state of the C$_{2p}$ orbital was found to emerge above the Conduction Band Minimum (CBM) upon the hydrogenation of CO₂* into HCOO* and become subsequently occupied upon the hydrogenation of HCOO* into H₂COO*. This orbital exhibits a π-bond character on both Zn₁-ZrO₂ and Zn₃-ZrO₂ surfaces, and contains substantial contribution from the d orbitals of Zr atoms bonded to the HCOO* intermediate but little contribution from the Zn atom bonded to the HCOO* intermediate (Supplementary Fig. 40). As a result, the center of this unoccupied C$_{2p}$ orbital band is positioned at 2.70 eV for the bri-HCOO$_{Zr,Zr}$* species on Zn₁-ZrO₂ and 3.27 eV for the bri-HCOO$_{Zn,Zr}$* species on Zn₃-ZrO₂ relative to the Fermi level. This leads to a greater difficulty of the bri-HCOO$_{Zn,Zr}$* species in acquiring electrons from negatively charged H* of H*-Zn$_{zr}$ than the bri-HCOO$_{Zr,Zr}$* species, explaining the much higher

barrier of bri-HCOO$_{Zn,Zr}$* hydrogenation reaction than of bri-HCOO$_{Zr,Zr}$* hydrogenation reaction. We calculated the C-O vibrational mode of bri-HCOO$_{Zn,Zr}$* and bri-HCOO$_{Zr,Zr}$* species to be 1563 and 1556 cm⁻¹, respectively. Thus, it is difficult to experimentally distinguish the bri-HCOO$_{Zn,Zr}$* and bri-HCOO$_{Zr,Zr}$* species using vibrational spectroscopy. Based on our results, previous argument of bri-HCOO$_{Zn,Zr}$* species as the active formate intermediate in the CO₂ hydrogenation to methanol reaction catalyzed by the Zn-O-Zr active site[30] is wrong.

### Reaction mechanism of CO hydrogenation reaction

ESR spectrum of Zn20Zr80 subjected to the CO hydrogenation reaction at 573 K without exposure to air exhibits substantially increased signals of O₂⁻ and F⁺ centers but barely changed signals of Zr³⁺ and Zn$_i$⁺ defects (Fig. 1d). Following a subsequent exposure to air, the signals of O₂⁻ and F⁺ centers are quenched whereas those of Zr³⁺ and Zn$_i$⁺ defects do not change (Supplementary Fig. 41), suggesting that the increased O₂⁻ and F⁺ centers on Zn20Zr80 subjected to the CO hydrogenation reaction at 573 K should probably locate on the Zn20Zr80 surface. Thus, the Zn20Zr80 surface is partially reduced during the CO hydrogenation reaction at 573 K to form O₂⁻ and F⁺ centers. CO hydrogenation reaction at 573 K catalyzed by Zn20Zr80 was operando characterized. An induction period of around 55 min was observed prior to the stable CH₃OH production (Fig. 6a), during which CO₂ and H₂O productions occurred, supporting the occurrence of partial surface reduction of Zn20Zr80. The vibrational features of bri-HCOO*, CH₃O*, CH₃OH* and OH* species (Supplementary Table 3) were observed to vary with the reaction time in the corresponding temporal

in situ DRIFTS (Fig. 6b), and their evolutions are shown in Fig. 6a. The OH* species initially decreases quickly and then slowly, and its consumption leads to the partial surface reduction of Zn20Zr80. The originally existing HCOO* species at 1590/1369 cm$^{-1}$ (denoted as bri-HCOO*-1590) initially increases quickly and then decreases slowly, while another new HCOO* species at 1561/1442 cm$^{-1}$ (denoted as bri-HCOO*-1561) emerges and keeps growing up to around 55 min. It is noteworthy that the formation of HCOO* species upon CO adsorption on Zn20Zr80 must involve one surface lattice oxygen site. The CH$_3$O* and CH$_3$OH* species emerge, grow and keep unchanged after around 15 min, demonstrating that both surface species reach the equilibrium states far before the HCOO* species and gaseous CH$_3$OH. Thus, the CH$_3$O* and CH$_3$OH* species should not be involved in the rate-limiting surface reaction step, while HCOO* hydrogenation reaction is the rate-limiting surface reaction.

Steady-state in situ DRIFTS spectra of Zn20Zr80-catalyzed CO hydrogenation reaction at various temperatures (Fig. 6c and Supplementary Fig. 42a) show that, with the temperature increasing, the bri-HCOO*-1590 species grows and reaches the maximum at 493 K and then weakens, while the bri-HCOO*-1561 species emerges at 493 K and then keeps growing. The evolutions of CH$_3$O* and CH$_3$OH* species are similar to that of bri-HCOO*-1561 species, not to that of bri-HCOO*-1590 species. The ESR spectra without exposure to air (Fig. 6d and Supplementary Fig. 42b) show that the additional O$_2^-$ and F$^+$ centers features emerge on the used Zn20Zr80 catalysts at 493 K and grows with the reaction temperature. The amount of paramagnetic centers in a sample is proportional to the area under its absorption curve that can be obtained by an integral treatment of the ESR spectrum[56,57], following which the intensities of O$_2^-$ and F$^+$ centers signals on Zn20Zr80 calcined in Ar at 773 K (I(O$_2^-$ + F$^+$ ESR signals)$_{Ar}$) and subsequently treated under 3 MPa CO + H$_2$ at different temperatures (I(O$_2^-$ + F$^+$ ESR signals)$_{CO+H2 \ reaction}$) were acquired. The intensity of vibrational feature of bri-HCOO*-1561 species on Zn20Zr80 treated under 3 MPa CO + H$_2$ at different temperatures and the CH$_3$OH formate rate of Zn20Zr80-catalyzed CO hydrogenation reaction at different temperatures were found proportional to the corresponding amount of created O$_2^-$ and F$^+$ centers on Zn20Zr80 surface, represented by I(O$_2^-$ + F$^+$ ESR signals)$_{CO+H2 \ reaction}$-I(O$_2^-$ + F$^+$ ESR signals)$_{Ar}$ (Fig. 6e). These results suggest that the active site of Zn20Zr80 in catalyzing the methanol production from CO hydrogenation is related with the in situ formed surface oxygen vacancy with the bri-HCOO*-1561 species as the active formate species, rather than the bri-HCOO*-1590 species on the stoichiometric surface.

Individual activation of CO and H$_2$ on Zn20Zr80 at 573 K was further studied (Supplementary Fig. 43). An exposure to 3 MPa CO forms bri-HCOO*-1590 and gaseous CO$_2$, and subsequent purging in Ar and heating in 3 MPa H$_2$ result in the formation of CH$_3$O* at the expense of bri-HCOO*. An exposure to 3 MPa H$_2$ demonstrates the hydrogenation of originally existing CO$_3$* species to bri-HCOO*-1590 and CH$_3$O*, and subsequent purging in Ar and heating in 3 MPa CO leads to the growth of bri-HCOO*-1590. These observations, on one hand, demonstrate that the bri-HCOO*-1590 species is capable of hydrogenating to produce CH$_3$OH, on the other hand, suggest that the reduction of Zn20Zr80 surface responsible for the formation of bri-HCOO*-1561 species is more facilitated under the syngas condition than under the individual CO or H$_2$ condition.

The Zn20Zr80 catalysts exclusively with the bri-HCOO*-1590 species or with the coexisting bri-HCOO*-1590 and bri-HCOO*-1561 species were prepared by the treatments in 3 MPa CO + H$_2$ (H$_2$:CO = 2) at 473 or 573 K for 60 min, respectively, following by cooling down to 303 K and purging in Ar (Supplementary Fig. 44a, b). Both formate species undergo the hydrogenation reaction in 3 MPa H$_2$ at 573 K. The hydrogenation reaction on the Zn20Zr80 catalyst exclusively with the bri-HCOO*-1590 species produces CO$_2$ but few CH$_3$OH, whereas the hydrogenation reaction on the Zn20Zr80 catalyst with the coexisting bri-HCOO*-1590 and bri-HCOO*-1561 species produces significantly more CH$_3$OH (Supplementary Fig. 44c, d). This further supports that the bri-HCOO*-1561 species is responsible for the CH$_3$OH production from CO hydrogenation over Zn20Zr80.

## Elementary surface reaction kinetics of CO hydrogenation reaction

The hydrogenation reaction on the Zn20Zr80 catalyst with the coexisting bri-HCOO*-1590 and bri-HCOO*-1561 specie in 3 MPa H$_2$ at various temperatures was characterized using temporal in situ DRIFTS spectra (Fig. 7a and Supplementary Fig. 45). The C-O stretch vibration peaks were peak-fitted (Supplementary Figs. 46–51) to give the evolutions of bri-HCOO*-1590 and bri-HCOO*-1561 species as a function of reaction time, respectively (Fig. 7b and Supplementary Fig. 53). Following the elementary surface reaction kinetic model proposed in Supplementary Fig. 52, the hydrogenation reaction of bri-HCOO*-1590 and bri-HCOO*-1561 was found to follow the first-order reaction kinetics (Fig. 7c, d), which is reasonable because the concentration of active H species during the hydrogenation reaction could be considered constant. Then the rate constants at various temperatures were calculated and consequently the Arrhenius plots (Fig. 7e) were plotted. The activation energy of bri-HCOO*-1590 and bri-HCOO*-1561 hydrogenation reactions was calculated as 121.6 ± 19 and 67.6 ± 9 kJ mol$^{-1}$, respectively. The rate for bri-HCOO*-1590 and bri-HCOO*-1561 hydrogenation reactions were also derived (Supplementary Fig. 54) and gives the activation energy of as 107.8 ± 17 and 68.9 ± 1 kJ mol$^{-1}$(Supplementary Fig. 55), respectively. Comparing the apparent activation energy of CO hydrogenation to methanol reaction catalyzed by Zn20Zr80 (54.5 kJ mol$^{-1}$), the elementary surface reaction kinetics clearly demonstrate that the bri-HCOO*-1561 species related with the surface defective sites in situ created on Zn20Zr80 is the active formate species for the CH$_3$OH production, whereas the bri-HCOO*-1590 species formed on the stoichiometric Zn20Zr80 surface is a spectator.

The bri-HCOO*-1590 species barely hydrogenates at 453 K (Supplementary Fig. 56), thus its increase on Zn20Zr80 in 3 MPa CO + H$_2$ at various temperatures below 453 K exclusively arises from its formation by CO hydrogenation, whose temporal in situ DRIFTS spectra were measured (Supplementary Fig. 57). Following the elementary surface reaction kinetic model proposed in Supplementary Fig. 58, the formation rates of bri-HCOO*-1590 species at different temperatures were acquired and used to plot the Arrhenius plot (Supplementary Fig. 59), whose results give the activation energy of bri-HCOO*-1590 formation reaction by CO hydrogenation as 34.3 ± 6 kJ mol$^{-1}$. A treatment of Zn20Zr80 in 3 MPa CO was also found capable of forming the bri-HCOO*-1590 species at the expense of originally-existing OH* species. Using the temporal in situ DRIFTS spectra (Supplementary Fig. 60) to measure the elementary surface reaction kinetics at various temperatures below 453 K (Supplementary Fig. 61), the activation energy of bri-HCOO*-1590 formation reaction and OH* consumption was calculated as 36.3 ± 3 and 37.5 ± 1 kJ mol$^{-1}$ (Supplementary Fig. 62). These results demonstrate that the OH* is capable of hydrogenating CO to produce the bri-HCOO*-1590 species on Zn20Zr80.

## Theoretical simulations of CO hydrogenation reaction

We firstly investigated the formation energy of an O vacancy (V$_O$) under the reaction conditions on both Zn$_1$-ZrO$_2$ and Zn$_3$-ZrO$_2$ surfaces, corresponding to a surface concentration of 0.0832 ML corresponding to stoichiometry t-ZrO$_2$ (110) surface. The most stable V$_O$ locates at the Zr-O-Zr site on Zn$_1$-ZrO$_2$ and Zn$_3$-ZrO$_2$ (Fig. 5a, c), whose formation energy are both 1.39 eV. However, by enlarging the surface to a 2×2 supercell that gives a V$_O$ concentration of 0.0208 ML, the formation energy of V$_O$ is then reduced to 0.91 eV on Zn$_1$-ZrO$_2$ and 0.30 eV on Zn$_3$-ZrO$_2$. These results suggest that the surface with aggregated Zn species is more likely to be reduced, which is consistent with the recent

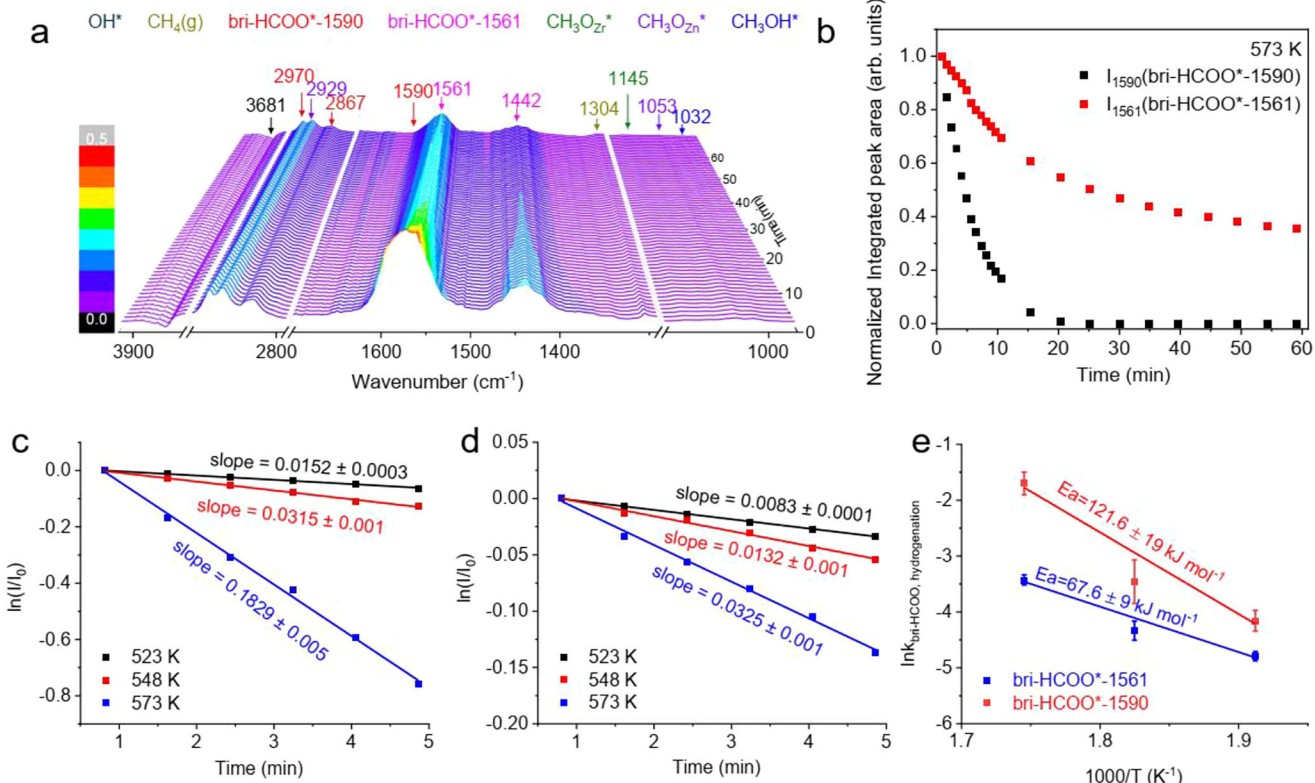

**Fig. 7 | Elementary surface reaction kinetics at high temperatures in CO hydrogenation reaction. a** Temporal in situ DRIFTS spectra of Zn20Zr80 (pretreated in 3 MPa CO + H₂, H₂:CO = 2, at 573 K for 60 min and purged in Ar) under 3 MPa H₂ at 523 K. **b** The corresponding normalized integrated peak area of observed species in Fig. 7a as a function of time on Zn20Zr80 catalyst at 573 K.

**c** First-order reaction kinetic of bri-HCOO*-1590 and **d**, bri-HCOO*-1561 hydrogenation reaction on Zn20Zr80 at different temperatures derived from panel (**b**), as well as Supplementary Fig. 53. **e** Arrhenius plots of bri-HCOO*-1590 and bri-HCOO*-1561 hydrogenation reactions derived from (**c, d**). The error bars in the figure represent the standard errors (SE) of the fitted values.

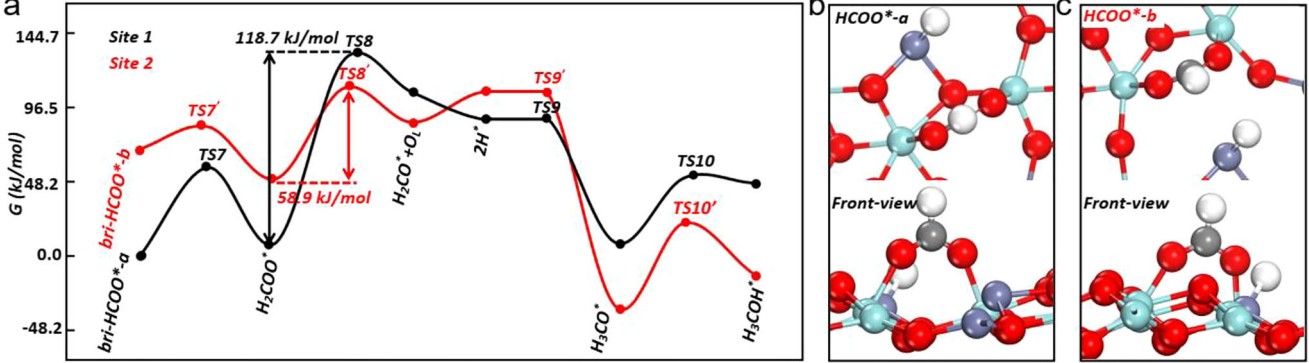

**Fig. 8 | Theoretical calculations of CO hydrogenation reaction. a** The energy profile of bri-HCOO* hydrogenation to CH₃OH at the most stable site (denoted as bri-HCOO*-a) and second most stable site (denoted as bri-HCOO*-b) of Zn₃-ZrO₂ surface. **b, c** The snapshots of bri-HCOO*-a and bri-HCOO*-b.

finding that the surface oxygen of ZnO will facilely oxidize CO into $CO_2$[58].

Then the possible pathways of CO hydrogenation into methanol were explored by firstly considering the possible pathways for surface hydroxyl reacting with adsorbed CO molecules (Supplementary Fig. 63). On both $Zn_1$-$ZrO_2$ and $Zn_3$-$ZrO_2$ surfaces, the most stable adsorption site for CO is at the Zr top site, where the CO molecule inserts into a Zr-O bond with the C atom to form an $OCO^*$ species, giving the adsorption energy of −24.1 and −31.8 kJ mol⁻¹, respectively. The optimal pathway of HCOO* formation is through a diffusion of hydrogen from hydroxyl group to the nearby Zn site, followed by reacting with $OCO^*$, to form a bri-$HCOO_{Zr,Zr}^*$ intermediate. The calculated barriers of H diffusion and bri-HCOO* formation is 60.8 and

183.3 kJ mol⁻¹ on $Zn_1$-$ZrO_2$, respectively, and 46.3 and 30.9 kJ mol⁻¹ on $Zn_3$-$ZrO_2$, respectively, while, as shown above, the experimentally-measured activation energies of hydroxyl consumption and bri-HCOO* formation during CO adsorption on the Zn20Zr80 catalyst respectively are 37.5 and 36.3 kJ mol⁻¹. Thus, the DFT calculation results indicate that the experimentally-observed HCOO* formation by CO reaction with OH groups should occur at the $Zn_n$ site of Zn20Zr80 surface rather than at the $Zn_1$ site. The resulting bri-$HCOO_{Zr,Zr}^*$ species on $Zn_3$-$ZrO_2$ (denoted as bri-HCOO*-a, Fig. 8a, b) hydrogenates with $H^*$-$Zn_{Zr}$ species via a barrier of 56.9 kJ mol⁻¹ to form $H_2COO^*$-a, and the C-O bond of $H_2COO^*$-a subsequently breaks to form $H_2CO^*$-a and a lattice oxygen via a barrier of 118.7 kJ mol⁻¹ (Fig. 8a and Supplementary Fig. 64), then $H_2CO^*$-a hydrogenates with $H^*$-$Zn_{Zr}$

barrierlessly, forming $CH_3O^*$-a, which further hydrogenates to produce $CH_3OH$ via a barrier of 42.5 kJ mol$^{-1}$. The overall barrier of bri-HCOO$^*$-a hydrogenation to $CH_3OH$ is 128 kJ mol$^{-1}$, similar to the experimentally-determined barrier of bri-HCOO$^*$-1590 hydrogenation on unreduced Zr20Zr80 surface under CO hydrogenation reaction condition.

We note that the most stable adsorption site of CO on $Zn_3$-$ZrO_2$ surface is also the most reducible site and hence the most stable formate intermediate (bri-HCOO$^*$-a) may not present under the reductive condition of CO hydrogenation reaction. Consequently, we investigated the hydrogenation activity of the second most stable bri-HCOO$_{Zr,Zr}^*$ species (denoted as bri-HCOO$^*$-b, Fig. 8c), 69.5 kJ mol$^{-1}$ less stable than the bri-HCOO$^*$-a species. The formation energy of $V_O$ (134.1 kJ mol$^{-1}$) is relatively high in the adopted model, but we found that the $Zn_{zr}$ site gains 1.1|e| with the presence of bri-HCOO$^*$-a, which, compared to the $Zn_{Zr}$ on clean surface, mimics a reduced $Zn_3$-$ZrO_2$ surface for bri-HCOO$^*$-b further hydrogenation reactions. As shown in Fig. 8a, on the $Zn_3$-$ZrO_2$ surface with the presence of bri-HCOO$^*$-a, bri-HCOO$^*$-b hydrogenates to produce methanol following the same reaction pathway as bri-HCOO$^*$-a, however, the barrier of C-O bond breaking in $H_2COO^*$-b to form $H_2CO^*$-b is only 58.9 kJ mol$^{-1}$, being much lower than that in $H_2COO^*$-a. Accordingly, the overall barrier of bri-HCOO$^*$-b hydrogenation into methanol is also 58.9 kJ mol$^{-1}$, similar to the experimentally-determined barrier of bri-HCOO$^*$-1561 hydrogenation on in situ partially-reduced Zr20Zr80 surface under CO hydrogenation reaction condition.

The bri-HCOO$^*$-a species and bri-HCOO-b species exhibit the calculated C-O vibrational mode at 1562 and 1549 cm$^{-1}$, respectively, and consequently correspond to the experimentally-observed bri-HCOO$^*$-1590 spectator and bri-HCOO$^*$-1561 active species on Zr20Zn80 for CO hydrogenation to methanol reaction. Thus, these DFT calculation results are consistent with the experimental results, supporting that the $Zn_n$ cluster bonded to an in situ formed -Zr-V$_O$-Zr- structure (-Zn-O-Zn(-O-Zr-V$_O$-Zr-)-O-Zr-O-Zr-) ($Zn_{n,Ov}$) on Zn20Zr80 is the active site for catalyzing the CO hydrogenation to methanol reaction and the associated bri-HCOO$^*$-1561 species is the active formate species whereas the unreduced $Zn_n$ cluster and $Zn_1$ site are not and the associated bri-HCOO$^*$-1590 are spectator. As shown above, the bri-HCOO$^*$ species at the unreduced $Zn_3$-$ZrO_2$ and $Zn_1$-$ZrO_2$ sites display similar C-O vibrational modes.

It is noteworthy that the CO hydrogenation to methanol reaction pathway at the in situ formed $Zn_{n,Ov}$ active site does not exist in the Zn20Zr80-catalyzed $CO_2$ hydrogenation reaction due to the absence of $Zn_{n,Ov}$ active site. This exemplifies the high sensitivity of surface structure of working catalyst to the reaction atmosphere and consequently the active site structure. It is also noteworthy that the bri-HCOO$^*$ species observed on Zn20Zr80 during $CO_2$ or CO hydrogenation to methanol reaction are with different structures and consequently formation ability and surface reactivity although they exhibit similar vibrational features. On one hand, both O atoms of bri-HCOO$^*$ species come from $CO_2$ for $CO_2$ adsorption while one O atom comes from CO and the other comes from Zn20Zr80 surface for CO adsorption; on the other hand, the local structure of surface site on Zn20Zr80 binding the bri-HCOO$^*$ species is different. The bri-HCOO$^*$ species is commonly observed as a surface intermediate in the vibrational spectra characterizing oxide catalysts for $CO_2$ or CO hydrogenation to methanol reaction, but its structures and surface reactivity need to be carefully identified in order to determine the role.

## $Zn_1$/m-$ZrO_2$ single atom catalyst

The above combined experimental and theoretical calculation results demonstrate that the $Zn_1$-single atom (-Zr-O-Zn-O-Zr-O-Zr-) on stoichiometric Zn20Zr80 and the $Zn_n$ cluster bonded to an in situ formed -Zr-V$_O$-Zr- structure (-Zn-O-Zn(-O-Zr-V$_O$-Zr-)-O-Zr-O-Zr-) on Zn20Zr80 are the active site for catalyzing the $CO_2$ and CO hydrogenation to

methanol reactions, respectively. Inspired by these findings, a $Zn_1$/m-$ZrO_2$ catalyst exclusively with the $Zn_1$ single atoms was prepared using a strong electrostatic adsorption method[59]. The Zn loading was measured as 1.48 at. % and the BET specific surface area was measured as 22.6 m$^2$/g. XRD pattern of $Zn_1$/m-$ZrO_2$ (Supplementary Fig. 65) matches the standard pattern of pure monoclinic $ZrO_2$ (JCPDS no.88-2390). The Zn K-edge FT-EXAFS spectrum (Fig. 9a, Supplementary Fig. 66 and Supplementary Table 8) reveals that $Zn_1$/m-$ZrO_2$ barely has the Zn-Zn coordination shell, and the HAADF-STEM and EDS mapping images and corresponding single-pixel line profiles (Fig. 9b, c and Supplementary Fig. 67) demonstrate the Zn atoms are singly dispersed on $ZrO_2$. The XPS spectrum shows that valence state of Zn in $Zn_1$/m-$ZrO_2$ is $Zn^{2+}$ (Supplementary Fig. 4), while the Zn K-edge XANES spectrum (Fig. 9d) demonstrates that the Zn K-pre-edge feature of $Zn_1$/m-$ZrO_2$ lies between those of Zn foil and ZnO powder.

Catalytic performance of $Zn_1$/m-$ZrO_2$ in the $CO_2$ and CO hydrogenation reactions was evaluated (Fig. 9e). The $Zn_1$/m-$ZrO_2$ catalyst is greatly more active than the Zn20Zr80 catalyst in catalyzing the $CO_2$ hydrogenation to methanol reaction but much less active in catalyzing the CO hydrogenation to methanol reaction, while the $CH_3OH$ selectivity of $CO_2$ or CO hydrogenation reaction catalyzed by $Zn_1$/m-$ZrO_2$ is similar to that by Zn20Zr80 at the same temperatures. The Arrhenius plots using $CH_3OH$ formation rate catalyzed by $Zn_1$/m-$ZrO_2$ (Supplementary Fig. 68) give an $E_a$ of 82.4 ± 3 kJ mol$^{-1}$ in the $CO_2$ hydrogenation reaction, similar to that catalyzed by Zn20Zr80 (78.9 ± 4 kJ mol$^{-1}$), and an $E_a$ of 84.2 ± 17 kJ mol$^{-1}$ in the CO hydrogenation reaction, much larger than that catalyzed by Zn20Zr80 (54.5 ± 6 kJ mol$^{-1}$). The used $Zn_1$/m-$ZrO_2$ catalyst exhibits a similar structure to the fresh one. These observations further confirm that the active site on the Zn20Zr80 surface for catalyzing the $CO_2$ hydrogenation to methanol is the $Zn_1$ atom but not for catalyzing the CO hydrogenation to methanol. However, a $Zn_1$/t-$ZrO_2$ catalyst exhibits much poorer catalytic performance in the $CO_2$ hydrogenation to methanol reaction (Supplementary Fig. 69), indicating that the $ZrO_2$ crystal phase strongly affect the catalytic performance of $Zn_1$/$ZrO_2$ catalysts in the $CO_2$ hydrogenation reaction.

The $H_2$ TPR profiles (Fig. 9f) show that $Zn_1$/m-$ZrO_2$ is less reducible than Zn20Zr80, consistent with the DFT finding that the oxygen vacancy at the $Zn_1$ site is more difficult to be created than at the $Zn_n$ site. This implies that the poor catalytic performance of $Zn_1$/m-$ZrO_2$ in the CO hydrogenation reaction probably results from its difficulty to be reduced and consequently the few surface oxygen vacancies. Reaction mechanisms of $Zn_1$/m-$ZrO_2$ catalyzed $CO_2$ or CO hydrogenation reaction were studied using in situ DRIFTS (Supplementary Figs. 70–72). In the $CO_2$ hydrogenation reaction, the low-barrier surface reaction pathway involving the key $H_2COO^*$ intermediate was identified at 303 K and above but the high-barrier surface reaction pathway was not, consistent with the exclusive presence of $Zn_1$ single atom sites on $Zn_1$/m-$ZrO_2$. In the CO hydrogenation reaction, the active bri-HCOO$^*$-1561 species was not observed, consistent with the few surface oxygen vacancies on $Zn_1$/m-$ZrO_2$.

## Discussion

In summary, using experimentally-measured elementary surface reaction kinetics to bridge the comprehensive experimental and theoretical calculation results, we unambiguously identify the active sites and associated reaction mechanisms of ZnO-$ZrO_2$ catalysts in the $CO_2$ or CO hydrogenation to methanol reaction. The $Zn_1$ single atom exclusively with the Zn-O-Zr (-Zr-O-Zn-O-Zr-) local structure and the $Zn_n$ clusters with both the Zn-O-Zr and Zn-O-Zn (-Zn-O-Zn-O-Zr-O-Zr-) local structures coexist on the Zn20Zr80 surface. In the $CO_2$ hydrogenation to methanol reaction, the $Zn_1$ single atom is the active site with the $CH_3O^*$ hydrogenation as the rate-limiting step. Interestingly, the experimentally-observed bri-HCOO$^*$ intermediate on the working Zn20Zr80 catalyst at the typical reaction temperatures is predominantly the bri-HCOO$_{Zn,Zr}^*$ spectator at the $Zn_n$ sites, but not the

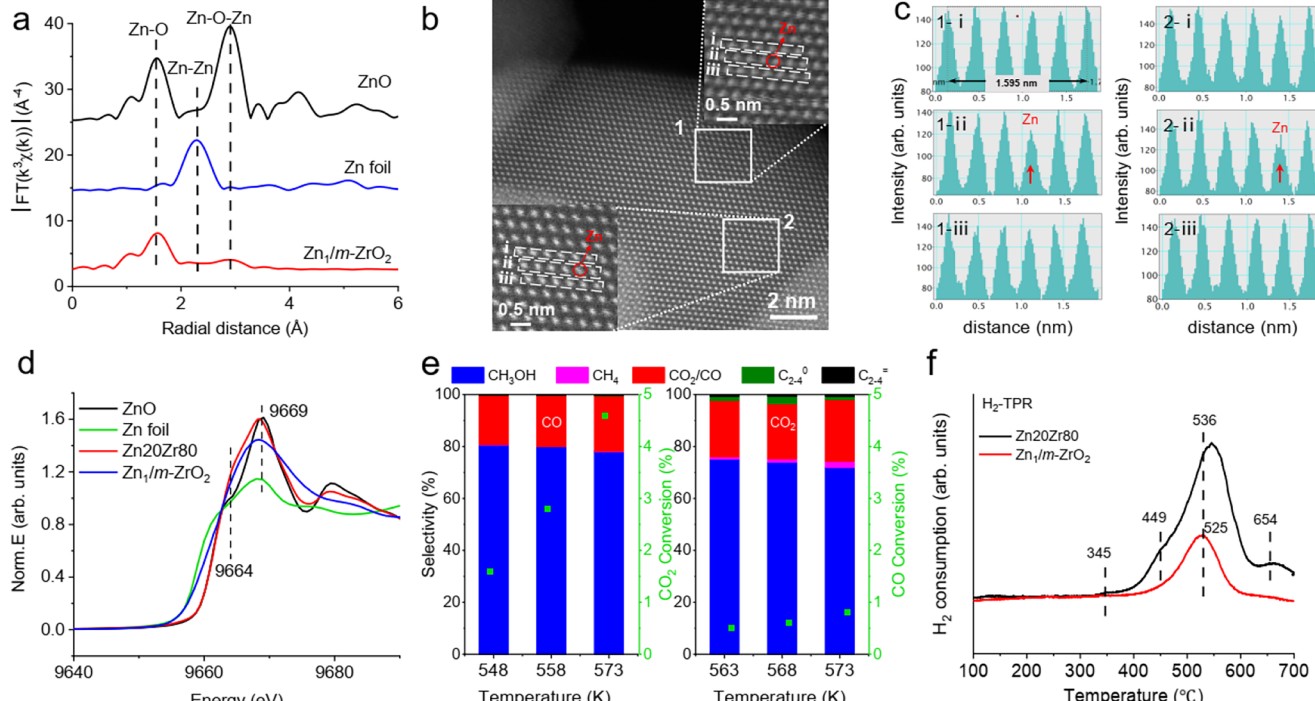

**Fig. 9 | Structural characterizations and catalytic performance of Zn₁/m-ZrO₂ catalyst. a** Fourier transforms of the $k^3$-weighted Zn K-edge EXAFS spectra of Zn₁/m-ZrO₂ catalyst and referring ZnO and Zn foil. **b** Atomically resolved STEM spectrum imaging the edge of the Zn₁/m-ZrO₂ nanoparticle. **c** The corresponding single-pixel line profiles across the atom rows marked by rectangles in panel 1-i, 1-ii, 1-iii and 2-i, 2-ii, 2-iii. The dark dots and low-intensity single-pixel line profiles correspond to Zn-containing atomic columns, marked with red circles and arrows. **d** Zn K-edge XANES spectra of Zn₁/m-ZrO₂ and Zn20Zr80 catalyst and referring ZnO and Zn foil. **e** Catalytic performance of Zn₁/m-ZrO₂ catalyst in $CO_2$ (catalyst mass: 300 mg; flow rate: 30 mL min⁻¹; $P$ = 3.0 MPa; H₂: $CO_2$ = 3) and CO hydrogenation (catalyst mass: 500 mg; flow rate: 16 mL min⁻¹; $P$ = 3.0 MPa; H₂: CO = 2). **f** H₂ Temperature Programmed Reduction (H₂ TPR) results of Zn20Zr80 and Zn₁/m-ZrO₂ catalyst.

active bri-HCOO$_{Zr,Zr}$* species at the Zn₁ sites whose coverage is too low to be observed. In the CO hydrogenation to methanol reaction, the surface oxygen vacancies are created at the Znₙ sites but seldom at the Zn₁ sites, and the Zn$_{n,Ov}$ site bonded to the in situ formed -Zr-V$_o$-Zr- structure (·Zn-O-Zn(-O-Zr-V$_o$-Zr-)-O-Zr-O-Zr-) is the active site with the bri-HCOO$_{Zr,Zr}$*-Zn$_{n,Ov}$ hydrogenation as the rate-limiting step. The experimentally-observed bri-HCOO$_{Zr,Zr}$*-Znₙ and bri-HCOO$_{Zr,Zr}$*-Zn₁ species on the working Zn20Zr80 catalyst is the spectator. The distinctly different active sites of Zn20Zr80 catalyst in the $CO_2$ and CO hydrogenation to methanol reactions illustrate high sensitivity of the active site structure of a catalyst to the reaction atmosphere. Nevertheless, the elementary surface reaction kinetics of working catalysts as a reliable correlation between experimental and theoretical calculation results provides a reliable methodology to unambiguously identify the active site of a catalyst in complex reactions.

## Methods
### Catalyst preparation
A co-precipitation method was adopted to synthesize the Zn20Zr80 composite oxide. Typically, the measured amounts of Zr(NO₃)₄•5H₂O (National Pharmaceutical Chemical Reagent Co., Ltd., ≥98.0%) and Zn(NO₃)₂•6H₂O (Chinese Pharmaceutical Chemical Reagent Co., Ltd., ≥ 99.0%) were dissolved in 100 mL deionized water (resistivity ≥ 18.25 MΩ·cm), to which another 100 mL aqueous solution containing 3.06 g (NH₄)₂CO₃ (Chinese Pharmaceutical Chemical Reagent Co., Ltd., ≥99.999%) was added dropwise at a rate of 3 mL/min under vigorous stirring at 70 °C. Then the mixture was stirred for an additional hour and aged for 3 h 70 °C, followed by cooling to room temperature. The resulting precipitate was filtered, washed with deionized water for three times, then dried at 80 °C for 16 h and finally calcined at 500 °C for 3 h in air to acquire the Zn20Zr80 sample. The same precipitation

method without an addition of Zn(NO₃)₂•6H₂O was also used to synthesize the m-ZrO₂ sample.

A Strong Electrostatic Adsorption (SEA) method[59] was adopted to synthesize the Zn₁/m-ZrO₂ catalyst using the m-ZrO₂ as the support. Typically, the measured amount of Zn(NO₃)₂•6H₂O was dissolved in 100 mL deionized water, after which the pH of the solution was adjusted to 10 with 25 wt.% ammonia solution. Then, the calculated amount of m-ZrO₂ with a fixed surface loading of 10³ m² L⁻¹ was mixed with 100 mL of the above Zn(NO₃)₂ aqueous solution at 25 °C. The resulting slurry was magnetically stirred for 3 h and filtrated. Then the recovered solid was dried at 80 °C for 12 h and finally calcined at 400 °C for 3 h in air to obtain the catalyst.

### Structural characterizations
Powder X-ray diffraction (XRD) patterns with a 2θ range of 10–80° were measured using a Philips X'Pert PROS diffractometer with a nickel-filtered Cu K α radiation source (0.15418 nm) operated with a voltage of 40 kV and a current of 50 mA, respectively. Transmission electron microscopy (TEM) and high-resolution transmission electron microscopy (HRTEM) images were obtained on Talos F200X and Titan Themis G2 60-300 microscopes, both operated at an accelerating voltage of 200 kV. Specific surface areas were determined using the Brunauer−Emmett−Teller (BET) method on a Micromeritics ASAP 2020 system after an adequate degassing of the samples at 573 K for 2 h in N₂. X-ray photoelectron spectroscopy (XPS) measurements were carried out on an ESCALAB 250 high performance electron spectrometer using monochromated Al Kα (hv = 1486.7 eV) as the excitation source. The C 1 s binding energy of adventitious carbon species was set to 284.8 eV to correct the likely charging effect. Sample compositions were analyzed by inductive coupled plasma atomic emission spectroscopy (ICP-AES) using an Opmita 7300DV spectrometer, during which

the sample was heated to 423 K in 1 mL hydrofluoric acid and then diluted with a determined volume of deionized water for the analysis.

ESR spectra were recorded at 140 K on a JEOL JESFA200 ESR spectrometer operating at 9.087 GHz with a microwave power of 0.998 mW, a modulation frequency of 100 kHz, and a modulation amplitude of 0.35 mT. The catalyst powders were placed in a high-pressure stainless-steel reactor tube equipped with two shut-off valves, allowing the sample to be isolated from ambient air. Following $CO_2$ or CO hydrogenation reaction under designated conditions, the reactor was cooled to 303 K in the reaction atmosphere. Then the tube was sealed and disconnected from the reactor and then transferred into an Ar-filled glovebox (Mikrouna Co. Ltd., China, with $O_2$ levels below 0.1 ppm) where 100 mg of the treated catalyst powders were carefully transferred into an ESR sample tube, which was subsequently sealed for ESR measurements.

Zn and Zr $K$-edge XAFS spectra were collected at the BL14W1 beamlines at the Shanghai Synchrotron Radiation Facility (SSRF) using Si (111) crystal monochromators. Prior to the measurements, the catalysts samples were compressed into thin sheets (1 cm in diameter) and sealed between Kapton films. The XAFS spectra were recorded at room temperature using a Bruker 5040 4-channel Silicon Drift Detector. Zn $K$-edge extended X-ray absorption fine structure (EXAFS) spectra were collected in fluorescence mode. The Zn and Zr $K$-edge XANES spectra of a specific sample exhibited negligible changes in the line-shape and peak position across repeated scans. Reference spectra of standard materials (Zn foil and ZnO) were obtained in transmission mode. Data reduction and analysis were performed using the Athena and Artemis software packages.

## Catalytic tests

Catalytic performance in the $CO_x$ hydrogenation reactions was evaluated on a high-pressure fixed-bed flow reactor custom-built by Xiamen Han De Engineering Co., Ltd. Typically, 1 g Zn20Zr80 catalyst with a grain size of 30–60 mesh (250–600 μm) or 0.3 g $Zn_1/m$-$ZrO_2$ catalyst diluted with 0.7 g of quartz sand was loaded in a Ti reactor with a 10 mm inner diameter and pretreated in flowing Ar (flow rate: 30 mL min$^{-1}$) at 773 K for Zn20Zr80 or 673 K for $Zn_1/m$-$ZrO_2$ for 2 h prior to reaction. After the reactor was cooled down to 303 K, the $CO_x$ + $H_2$ flow (Total pressure: 3 MPa; $H_2$: $CO_2$ = 3:1 or $H_2$: CO = 2:1) containing Ar with a concentration of 3% as an internal standard for the calculation of $CO_x$ conversion was introduced with a total flow rate of 30 mL min$^{-1}$. The composition of effluent gas reaction was analyzed with an online gas chromatograph (Ruimin GC2060, Shanghai) equipped with both a thermal conductivity detector (TCD) and a flame ionization detector (FID). And the selectivity was calculated on a molar carbon basis. The activity and selectivity of the catalytic reaction were calculated as shown in Eqs. (1)–(6) below, in which $X_i$ represents conversion of substance $i$, $S_i$ represents selectivity of substance $i$, $x$ represents carbon chain length of $C_{2+}$ components and $n_i$ represents moles of substance $i$:

$$X_{CO_x}(\%) = \frac{n_{CO\ or\ CO_{2\,produced}} + n_{CH_3OH_{produced}} + n_{CH_{4\,produced}} + 2n_{DME_{produced}} + \Sigma x n_{C^0_{2-4\,produced}}}{n_{CO_{x\,input}}} \times 100\% \quad (1)$$

$$S_{CH_3OH}(\%) = \frac{n_{CH_3OH_{produced}}}{n_{CO\ or\ CO_{2\,produced}} + n_{CH_3OH_{produced}} + n_{CH_{4\,produced}} + 2n_{DME_{produced}} + \Sigma x n_{C^0_{2-4\,produced}}} \times 100\% \quad (2)$$

$$S_{CO\ or\ CO_2}(\%) = \frac{n_{CO\ or\ CO_{2\,produced}}}{n_{CO\ or\ CO_{2\,produced}} + n_{CH_3OH_{produced}} + n_{CH_{4\,produced}} + 2n_{DME_{produced}} + \Sigma x n_{C^0_{2-4\,produced}}} \times 100\% \quad (3)$$

$$S_{CH_4}(\%) = \frac{n_{CH_{4\,produced}}}{n_{CO\ or\ CO_{2\,produced}} + n_{CH_3OH_{produced}} + n_{CH_{4\,produced}} + 2n_{DME_{produced}} + \Sigma x n_{C^0_{2-4\,produced}}} \times 100\% \quad (4)$$

$$S_{DME}(\%) = \frac{2n_{DME_{produced}}}{n_{CO\ or\ CO_{2\,produced}} + n_{CH_3OH_{produced}} + n_{CH_{4\,produced}} + 2n_{DME_{produced}} + \Sigma x n_{C^0_{2-4\,produced}}} \times 100\% \quad (5)$$

$$S_{C_{2-4}}(\%) = \frac{\Sigma x n_{C^0_{2-4\,produced}}}{n_{CO\ or\ CO_{2\,produced}} + n_{CH_3OH_{produced}} + n_{CH_{4\,produced}} + 2n_{DME_{produced}} + \Sigma x n_{C^0_{2-4\,produced}}} \times 100\% \quad (6)$$

## In situ and temporal DRIFTS and online MS measurements

In situ and temporal high-pressure DRIFTS measurements were carried out using a Nicolet IS50 FT-IR spectrometer equipped with an in situ high pressure DRIFTS reaction cell (Harrick Scientifc Products, INC) and connected with a mass spectrometer (Hiden Analytical, YQ074159).DRIFTS spectra were collected using a MCT/A detector with 128 scans per spectrum and a resolution of 4 cm$^{-1}$. The catalyst sample was placed on the reaction cell stage and pretreated under flowing Ar (flow rate, 30 mL min$^{-1}$) at 773 K for 2 hours. After cooling to the desired reaction temperature, the background spectrum was recorded. The feed gas was then switched to a mixture of 3 MPa $CO_x$ + $H_2$ ($H_2$: $CO_2$ = 3:1 or $H_2$: CO = 2:1) with a flow rate of 30 mL min$^{-1}$ and then the catalyst was heated to the target temperature at a rate of 10 K min$^{-1}$, held for 1 h, and subsequently cooled to 303 K before being purged with Ar. During this entire process, temporal in situ DRIFTS spectra were continuously recorded. The peak fitting processes of temporal in situ DRIFTS spectra were carried out using the "Peak Resolve" function of the "Omnic" software integrated with the infrared spectrometer. Typically, a set of original temporal in situ DRIFTS spectra within a fixed wavenumber range for a specific vibrational mode was loaded and automatically processed by selecting a "linear" baseline correction, a "Gaussian/Lorentzian" peak shape, a position constrained within a reasonable range and a fixed full width at half maximum (FWHM) of the peak of the same species to acquire the fitted spectra.

$CH_3OH$ chemisorption measurements were also performed on the Nicolet IS50 FT-IR spectrometer. The catalyst sample on the reaction cell stage was firstly pretreated in Ar at 773 K for 2 h and then cooled to 303 K whose spectra were taken as the background spectra. Then the reactor cell was evacuated by a mechanical pump, filled with the $CH_3OH$ vapor for 30 min, and evacuated again. Then the DRIFTS spectra were recorded.

Sodium formate (Sigma Aldrich) was used as the formate source and deposited on the catalyst with weight loadings of 1%, 2%, and 5% using the wetness impregnation method. The catalyst was dried at 353 K overnight. The IR background was measured using the catalyst only at 303 K with Ar (30 ml/min). After that, the catalyst with sodium formate was loaded into the DRIFTS cell and baked at 353 K with Ar for 1 hour. Then, the spectra were taken after the DRIFTS cell was cooled down to room temperature. By integrating the peak area of the formate species, the calibration curves were obtained.

## Theoretical calculations

**Stochastic surface walking global optimization.** The Stochastic Surface Walking (SSW) algorithm incorporates an automated climbing mechanism that drives a system from a local minimum to a high-energy configuration along a randomly chosen direction, which is inherited from the bias-potential driven constrained-Broyden-dimer (BP-CBD) method originally developed for transition state searches. The SSW algorithm has proven effective in exploring complex potential energy surfaces (PES) and predicting stable structures. In each SSW step, a modified potential energy surface $V_{mod}$ is employed to guide the system from the current minimum $R_t^0$ to a high energy configuration $R_t^H$. This is achieved by sequentially adding a series of bias Gaussian potentials $v_n$ (n is the index of the bias potential,

$n = 1, 2, …, H$) along the corresponding direction $N_t^n$ as shown in Eq. (7), where $R_t$ denotes the coordinate vector of the structure, and $V_{real}$ corresponds to the original, unmodified potential energy surface; $R_t^n$ represents the $n$-th local minimum encountered along the climbing trajectory on the modified PES, which is progressively shaped by the addition of $n$ Gaussian bias potentials. Once the system reaches the high-energy configuration $R_t^H$, all bias potentials are removed, and a local geometry optimization is performed to relax the structure to a nearby minimum. At the end of each SSW step, a Metropolis Monte Carlo scheme is employed to determine whether the newly obtained minimum is accepted or rejected.

$$V_{mod} = V_{real} + \sum_{n=1}^{NG} v_n = V_{real} + \sum_{n=1}^{NG} w_n * \exp[-((R_t - R_t^n) \cdot N_t^n)^2/(2 \times ds^2)]$$

(7)

**Neural Network potential energy surface.** In this work, we utilized the NN scheme that firstly introduced by Behler and Parrinello for constructing the global NN PES. The total energy $E^{tot}$ of a structure is decomposed and described as a linear combination of atomic energy $E_i$ from the output of NN, where six power-type structure descriptors (PTSDs), namely $S^1$ to $S^6$, are utilized as the input of the network. These descriptors describe the geometrical environment of atom in the following mathematic forms:

$$E^{tot} = \sum_i E_i$$

(8)

$$f_c(R_{ij}) = \begin{cases} 0.5 \times tanh^3\left[1 - \frac{r_{ij}}{r_c}\right], for\ r_{ij} \leq r_c \\ 0, for\ r_{ij} > r_c \end{cases}$$

(9)

$$R^n(r_{ij}) = r_{ij}^n \cdot f_c(r_{ij})$$

(10)

$$S_i^1 = \sum_{j \neq i} R^n(r_{ij})$$

(11)

$$S_i^2 = \left[\sum_{m=-L}^{L} | \sum GU_2|^2\right]^{\frac{1}{2}} = \left[\sum_{m=-L}^{L} | \sum_{j \neq i} R^n(r_{ij}) Y_{Lm}(r_{ij})|^2\right]^{\frac{1}{2}}$$

(12)

$$S_i^3 = 2^{1-\zeta} \sum GU_3 = 2^{1-\zeta} \sum_{j,k \neq i} (1 + \lambda cos\theta_{ijk})^\zeta \cdot R^n(r_{ij}) \cdot R^m(r_{ik})$$

(13)

$$S_i^4 = 2^{1-\zeta} \sum GU_4 = 2^{1-\zeta} \sum_{j,k \neq i} (1 + \lambda cos\theta_{ijk})^\zeta \cdot R^n(r_{ij}) \cdot R^m(r_{ik}) \cdot R^p(r_{ik})$$

(14)

$$S_i^5 = \left[\sum_{m=-L}^{L} | \sum GU_5|^2\right]^{\frac{1}{2}}$$

$$= \left[\sum_{m=-L}^{L} | \sum_{j,k \neq i} R^n(r_{ij}) \cdot R^m(r_{ik}) \cdot R^p(r_{ik}) \cdot (Y_{Lm}(r_{ij}) + Y_{Lm}(r_{ik})|^2\right]^{\frac{1}{2}}$$

(15)

$$S_i^6 = 2^{1-\zeta} \sum GU_6 = 2^{1-\zeta} \sum_{j,k,l \neq i} (1 + \lambda cos\theta_{ijk})^\zeta \cdot R^n(r_{ij}) \cdot R^m(r_{ik}) \cdot R^p(r_{il})$$

(16)

The dataset for training the neural network potential energy surface (NN PES) is constructed through an iterative process. The initial dataset is generated from density functional theory (DFT)-based SSW simulations, while subsequent data are obtained from SSW explorations driven by the evolving NN PES. These SSW simulations cover a wide range of compositions and morphologies—including bulk, layered, and cluster structures—with varying atomic compositions and unit cell sizes. In total, the simulations sampled over $10^7$ structures on the PES, from which 58,883 representative structures were selected to form the final global dataset, all of which were recalculated using high-accuracy DFT. To ensure high accuracy of the PES, we employed a comprehensive set of PTSDs, comprising 493 descriptors per element: 282 two-body, 171 three-body, and 40 four-body terms. Accordingly, a large neural network architecture was adopted, consisting of two hidden layers with a 100-80-80-6 structure.

**SSW-NN calculations.** Traditional density functional theory (DFT) calculations are often impractical for the global optimization of complex systems due to their prohibitively high computational cost. To overcome this limitation, we employed SSW global optimization based on NN PES to resolve the structure of the ZnZrO slab model, achieving performance 3 to 4 orders of magnitude faster than conventional DFT. The application of the SSW-NN method follows six main steps: (1) Generating a global dataset from the SSW trajectories and computing reference energies and forces using DFT; (2) Training the NN PES on this dataset; (3) Benchmarking the NN predictions against DFT results for selected structures from SSW trajectories, and augmenting the dataset with new configurations for retraining; (4) Iterating steps (1) through (3) until the PES deviation is sufficiently small (typically below 10 meV/atom); (5) Conducting SSW-based global optimization using the converged NN PES for the target system; (6) Recalculating the energies of key structures using high-accuracy DFT. In this study, the NN PES developed for the ZnZrO system achieves good agreement with DFT, with energy and force deviations of 3.696 meV/atom and 0.117 eV/Å, respectively.

**The Gibbs free energy.** For solids, the free energy is approximated by its DFT total energy. For molecules, the free energy is calculated as Eq. (17), including the DFT total energy [$X$], the zero-point-energy (ZPE) and the thermodynamic correction terms. The enthalpy changes and entropies for CO, $CO_2$, $H_2$, and $H_2O$ under reaction and standard conditions, as well as for $CH_3OH$ under the standard condition, are obtained from the NIST-JANAF Thermochemical Tables[60] and the enthalpy change and entropy of $CH_3OH$ under the reaction condition are calculated from DFT[61] (Supplementary Table 9).

$$G[X](p, T) = E[X] + ZPE[X] + [\boldsymbol{H}[X](p^0, T) - \boldsymbol{H}[X](p^0, 0K) - TS[X](p^0, T) + k_BTlnP/P^0]$$

(17)

**Surfaces search.** We employed Stochastic Surface Walking global optimization[62–64] with global neural network (G-NN) potential method (SSW-NN)[65,66] to characterize the different Zn concentration doping on the $t$-$ZrO_2$ (101) surface structures. The global minimums (GM) of active site structures were determined by the SSW-NN method, as implemented in the large-scale atomic simulation with neural network potential (LASP) code[54]. The Zn−Zr−O PES describes the global neural network (G-NN) potential that was obtained by self-learning the dataset produced from SSW global optimization. The G-NN potential is available from the LASP project (www.lasphub.com). More details on SSW-NN and the Zn−Zr−O G-NN potential are provided in the Supporting Information.

In the SSW-NN simulation, the $t$-$ZrO_2$ (101) surface is modeled by a five-layer slab, where the bottom three layers were fixed at the bulk-truncated position and the top two layers were allowed to relax.

Considering that the $t$-$ZrO_2$ (101) surface supercell contains 12 Zr atoms, the doping we considered corresponds to 8.3% (1 Zr substituted on surface), 16.6% (2 Zr substituted on surface), and 25.0% (3 Zr substituted on surface) Zn atoms on the surface. The supercell had a dimension of 12.8 Å × 10.9 Å × 30.0 Å. And the relative stability is given by Eq. (18):

$$\Delta E_r = [E(Zn_3ZrO_2) + 2 \times E(slab - ZrO_2) - 3 \times E(Zn_1ZrO_2)]/3 \quad (18)$$

**DFT calculations.** The reaction pathways on the ZnZrO active site were calculated using first-principles DFT as implemented in the Vienna Ab initio Simulation Package (VASP)[67]. The exchange–correlation function utilized was GGA-PBE[68]. The electron–ion interaction was represented by the projector-augmented-wave pseudopotential[69,70], and the kinetics energy cutoff utilized was 450 eV. The first Brillion zone k-point sampling utilizes only the gamma-point since the supercell was rather large, which was shown to provide converged energetics. The energy and force criterion for convergence of the electron density and structure optimization were set at $10^{-5}$ eV and 0.05 eV/Å, respectively. All of the reaction TSs were determined using the constrained Broyden dimer (CBD) method[64,71], and the TS was further confirmed by the vibrational frequency analysis and the geometry extrapolation to the neighboring minima.

**Microkinetic simulation.** Microkinetic simulation was performed to evaluate the theoretical apparent activation energy. In the simulation, the pressure of $CO_2$ and $H_2$ were fixed at the reaction condition (3 MPa, $H_2$: $CO_2$ = 3) to simulate a fluidized-bed catalytic reactor during the whole process. The selected temperatures were 548, 558 and 573 K.

## Data availability
The data supporting the findings of the study are available within the paper and its Supplementary Information. Source data are provided with this paper.

## Code availability
The software code of LASP and Zn-Zr-O G-NN potential used within the article are available from the corresponding author (C.S.) upon request or from the website http://www.lasphub.com.

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

## Acknowledgements

This work was financially supported by the National Key R&D Program of MOST (2022YFA1504601, 2018YFA0208600), the Strategic Priority Research Program of the Chinese Academy of Sciences (XDB0450102), the National Natural Science Foundation of China (92145302, 91745202, 22122301, 92061112, 22033003, 22250710677), the Fundamental Research Funds for the Central Universities (20720220008, 20720220011), the Changjiang Scholars Program of Ministry of Education of China, the University of Science and Technology of China (KY2060000176), and the Tencent Foundation for XPLORER PRIZE. This work was partially carried out at the Instruments Center for Physical Science, University of Science and Technology of China.

## Author contributions

J.D. and Y.P. contributed equally to the work. W.H. conceived and supervised the project, oversaw all data analysis and discussion. C.S. supervised the theoretical simulations. project J.D. performed the experiments and analyzed the data. Y.P. and C.S. performed the theoretical simulations and analyzed the data. W.X., D.W. and Q.N, assisted with the experiments. Z.X. and Z.J. assisted with the EXAFS data analysis. Z.L. assisted with the DFT calculation. W.H., J.D., C.S. and Y.P. prepared the manuscript. All the authors discussed the results and commented on the manuscripts.

## Competing interests

The authors declare no competing interests.
