## [Transparent Peer Review file · Nature Communications]

Distinctly different active sites of ZnO-ZrO₂ catalysts in CO₂ and CO hydrogenation to methanol reactions

Corresponding Author: Professor Weixin Huang

Version 0:

Reviewer comments:

Reviewer #1

(Remarks to the Author)

Let me start by saying that I like the very detailed, comprehensive and complementary study with the aim to shed light on the active sites of recently proposed ZnO-ZrO₂ catalysts for the hydrogenation of both CO and CO₂ to methanol. This is a very timely study that has been performed with great care using a range of cutting edge experimental and theoretical methods. In general, I would think that this study deserves publication in Nature Communications, but there are a few issues that need to be addressed before final acceptance.

(1) Let me start with some of the main issues. Figure 2 shows the performance of ZnO-ZrO₂ for CO and CO₂ hydrogenation to methanol. The CO₂ conversion seems very high at 573 K, so I assume that this must be very close to the equilibrium? It would be good to indicate the equilibrium conversion at that temperature. Importantly, the Arrhenius plots thus include data that is very close to equilibrium (can also be seen from the bending of the high temperature point). I would therefore tend to be very careful with the interpretation of the apparent activation barriers. How would the slope look like if only differential conversion data would be included (e.g. only low T data points, or those measured at differential reaction conditions). It looks to me as if the apparent activation barrier could be substantially higher.

(2) On that note, at larger CO₂ (or CO) conversion, there will always be substantial water-gas-shift or rWGS activity occurring. So, for example CO₂ might be converted to CO which than is converted to methanol. It would be good to at least mention this dilemma at higher conversions.

(3) The authors observe from their studies that “However, the activation energy of bri-HCOO* hydrogenation reaction is much larger than the apparent activation energy of CO₂ hydrogenation to methanol reaction (78.9 kJ mol⁻¹), therefore, the high-barrier surface reaction pathway is not likely to be responsible for the methanol synthesis from CO₂ hydrogenation catalyzed by Zn₂₀Zr₈₀ catalyst ... “ and that “... 62.3 and 69.2 kJ mol⁻¹, slightly lower than the apparent activation energy of CO₂ hydrogenation to methanol (78.9 kJ mol⁻¹). Thus, the low-barrier surface reaction pathway is predominantly responsible for the methanol synthesis from CO₂ hydrogenation catalyzed by the Zn₂₀Zr₈₀ catalyst.” I am not sure that comparing those observed activation barriers with those in Figure 2 is totally fair (see comment above), as this might be also largely due to differences in measurement protocols. I would think, however, that the trends towards e.g. lower activation barriers for CO hydrogenation would still hold.

(3) Concerning the DFT calculations, there are no barriers reported for the H₂ dissociation, e.g. over the Zn-O pair. For metal-oxides this might be an important rate determining step and should be considered.

(4) The calculated gibbs free energy at 573K is barely negative, so this must be close to equilibrium (see comment above). Does the DFT method give the correct ΔH for CO and CO₂ hydrogenation (e.g. for CO₂ + H₂ → methanol a ΔH of -49 kJ/mol)?

(5) Experimentally there is a difference observed between bri-HCOO at 1590 and 1561 (the latter involving presumably oxygen from the lattice). Does DFT reproduce this difference of about 30 cm⁻¹ also when comparing bri-HCOO on Zr-Zn and

with a lattice O, at least qualitatively? I do understand though that the total numbers might differ by more than 30 cm⁻¹ between experiment and theory.

(6) Figure 4 shows temporal DRIFTS and then the time evolution of certain peaks (in b and f). However, I do not see the correlation of some of the peaks, e.g. the 2955 in b) is absent in a). Likewise, I do fail to see the 2896 and 2925 peaks shown in f) in figure e).

(7) The authors use the PBE functional for their DFT part, but seem not to include dispersion interaction (e.g. through D3). Since this can usually be done with little extra computational cost and may actually be important for larger adsorbates such as H₂COO, I wonder why this has not been included and what the effect would be?

(8) In general, the SI has been done with great care. It would be good to add the different contributions coming from the simulations in Table S4 though. E.g. instead of only giving ΔE and ΔG , I would prefer all calculated quantities, e.g. ΔE , ZPE, ΔS . That way an interested reader can easily comprehend how the numbers have been derived. Importantly, that would also allow anyone to replot the ΔG diagram at other temperatures, if of interest.

(9) I would recommend to only use one unit throughout the manuscript. For example in Figure 5, ΔG is given in eV, but (experimental and theoretical) Arrhenius plots are given in kJ/mol.

Reviewer #2

(Remarks to the Author)

This article reports on an in-depth study on the comparison between CO and CO₂ hydrogenation over Zn/ZrO₂ systems for the selective production of methanol. The authors propose two different active systems for both reaction and also elucidate the reaction mechanism by a combination of theoretical calculations and spectroscopy measurements. The article is of clear relevance to the field and also important for the further progression to obtain more selective, active and ultimately also stable catalyst materials for this important conversions process.

Unfortunately, I find the experimental basis on which the conclusions is based not that strong as a deeper analysis of the experimental data, as reported in the main text and the SI, is required. The spectroscopy data reported reveal that the "operando mode" only holds for the IR spectroscopy data where a clear attempt is done by the authors to correlate the different species observed with the selectivity and activity patterns changing with time on stream. As a side comment, I have to note here that in the SI the fitting of some of the data with a very steep background (Figures S26-S28) is not that accurate and certainly prone to errors. Having stated this, it would be worthwhile to further corroborate on this quantitative/semi-quantitative analysis.

I have to disagree with the authors on the notion that Figure 6 would suggest that an "operando/in-situ mode" is achieved; as stated in the figure caption. I also would not call the EPR data even quasi in-situ. The samples are just "trapped" from the reaction environment and cooled down to be measured at the low temperature required to measure the EPR spectra (e.g., 140 K). The entire EPR data have to be investigated in much more depth. Here, the authors have first to show the entire range from 0 to 6000 G (or something like this) and focus on the different signals observed. I am curious to see if there are other regions of interests and what the impurities are in the support material. Regarding the identification of the different signals, it is important to plot their intensity as a function of the measurement temperature to see if they have Curie-Weiss law characteristics and are paramagnetically isolated sites. Also some EPR simulations would be here advantageous. The authors argue that the addition of O₂ leads to a disappearance of the signal but that may not wonder as O₂ is a paramagnetic molecule itself. Furthermore, the authors have to further analyze in depth - via a systematic analysis of the different reaction and measurement parameters - the EPR spectra a what can be learned from this. These data as they are not in-situ or operando makes that only static information can be obtained. I am also a bit surprised that the plot in Figure 6e is with an x-axis which is relative (not absolute) and only ranging between 1 and 2, while the two y-axis have a 0 as the lowest value. This plot is not understandable to me and the question is how accurate this is and how it is exactly obtained (knowing the difficulties associated with the EPR measurements and the different species observed).

The comment on the operando/in-situ mode of operation for EPR also holds for the measured/reported XAS (EXAFS and XANES) as well as the HRTEM as they also are taken under "fixed" conditions, far from the real reaction conditions. Hence, the IR spectra are the only one which provides something about the reaction mechanism, but not that much can be stated about the active sites in the two set of reactions/systems.

The comments above brings me to the more general question that one may question why CO mostly formed from CO₂ requires such a different reaction site in this process relative to each other and that one expect a first reaction between CO₂ towards CO and then further from CO to methanol. I can expect that there two routes and hence two active phases, but the question is then if they are both not interrelated and why would site isolated be needed for one route and a cluster-type active site for the other route. This is not clear when reading the paper.

Summarizing, this article requires a lot of rework and additional data as well as data analysis to convince me as referee (and most probably also later on the reader of the work) that the conclusions are indeed correct/valuable. As a result, I propose major revision for this work before it can be accepted in Nature Communications.

Version 1:

Reviewer comments:

Reviewer #1

(Remarks to the Author)

The authors improved the manuscript considerably, addressing all my concerns. I therefore recommend to accept the manuscript for publication.

Reviewer #2

(Remarks to the Author)

I have now read the revised version of the paper, as well as the rebuttal letter. The authors have done a real effort to address my initial comments and also have made sure that a more quantitative approach is taken into the analysis of the spectroscopic data and also have nuanced where needed the aspect of the operando/in-situ/fixed reaction environment conditions thereby showing the strength and weaknesses of their analytical approaches taken. As far as I can judge, I do believe the article is now of sufficient quality and novelty to be published in Nature Communications and hence I propose it to be accepted as such.

Author Reply to Reviewer #1's Comments

Let me start by saying that I like the very detailed, comprehensive and complementary study
with the aim to shed light on the active sites of recently proposed ZnO-ZrO₂ catalysts for
the hydrogenation of both CO and CO₂ to methanol. This is a very timely study that has
been performed with great care using a range of cutting edge experimental and theoretical
methods. In general, I would think that this study deserves publication in Nature
Communications, but there are a few issues that need to be addressed before final
acceptance.

**Response:** We appreciate the reviewer's effort in reviewing our manuscript, positive
recommendation and insightful comments very much. We have seriously considered the
comments and revised the manuscript accordingly. We hope that the revised manuscript
will be suitable for the publication.

(1) Let me start with some of the main issues. Figure 2 shows the performance of ZnO-ZrO₂
for CO and CO₂ hydrogenation to methanol. The CO₂ conversion seems very high at 573 K,
so I assume that this must be very close to the equilibrium? It would be good to indicate the
equilibrium conversion at that temperature. Importantly, the Arrhenius plots thus include
data that is very close to equilibrium (can also be seen from the bending of the high
temperature point). I would therefore tend to be very careful with the interpretation of the
apparent activation barriers. How would the slope look like if only differential conversion
data would be included (e.g. only low T data points, or those measured at differential
reaction conditions). It looks to me as if the apparent activation barrier could be
substantially higher.

**Response:** We appreciate the reviewer's insightful comments very much. The CO₂
conversion at 573 K under the adopted reaction condition catalyzed by our Zn₂₀Zr₈₀
catalyst was 9.1%, very close to the equilibrium CO₂ conversion of 9.3%. We agree with the
reviewer that such data are not suitable to be used for reaction kinetic analysis. Therefore,
during the revision, we re-evaluated the catalytic reaction at 523, 533 and 548 K using 600
28 mg Zn₂₀Zr₈₀ catalyst diluted with 400 mg SiC to acquire reliable data for reaction kinetic
analysis. The results are summarized in Table R1:

Table R1. Catalytic performance of Zn₂₀Zr₈₀ catalyst in CO₂ hydrogenation reaction (3 MPa
with H₂; CO₂ = 3; flow rate: 30 mL min⁻¹; catalyst mass: 600 mg Zn₂₀Zr₈₀ diluted with 400
32 mg SiC).

Temperature (K)	CO ₂ Conversion (%)	Selectivity (%)			
		CH ₃ OH	CO	CH ₄	C ₂₊ ⁰
523	1.07	89.8	8.1	1.2	0.4
533	1.59	88.2	10.7	0.4	0.3
548	2.89	88.0	11.4	0.3	0.3

The corresponding Arrhenius plot in Figure R1 gives an activation barrier for CO₂

hydrogenation to methanol of $81.9 \pm 1 \text{ kJ mol}^{-1}$, slightly larger than the previous data ($78.9 \pm$
4 kJ mol^{-1}). The value is also similar to previously-reported apparent activation energies of
CO_2 hydrogenation to methanol reaction catalyzed by ZnO-ZrO_2 catalysts (81.2 kJ mol^{-1}
(*Chin. J. Catal.* 45, 162-173 (2023)) and 84.9 kJ mol^{-1} (*Angew. Chem., Int. Ed.* 63, e202316874
(2024))).

Figure R1. Arrhenius plot of CH_3OH formation reactions over $\text{Zn}_{20}\text{Zr}_{80}$ catalyst. Reaction
condition: 3 MPa with H_2 : $\text{CO}_2 = 3$; flow rate: 30 mL min^{-1} ; catalyst mass: 600 mg $\text{Zn}_{20}\text{Zr}_{80}$
diluted with 400 mg SiC .

Action

In the revised manuscript, we have added the above Table R1 as Supplementary Table 2
and replaced the original Figure 2b with Figure R1 and described the relevant results as the
following:

"The CO_2 conversion of 9.1% at 573 K is close to the equilibrium value (around 9.3%) under
the adopted reaction condition⁴⁹. We then measured the catalytic performance within the
kinetics-controlled range (Supplementary Table 2). The derived Arrhenius plots (Fig. 2b and
Supplementary Fig.11) calculated from the CH_3OH formation and CO_2 reaction rates give
similar apparent activation energy(E_a) of 81.9 ± 1 and $85.1 \pm 1 \text{ kJ mol}^{-1}$, respectively, consistent
with the literature results^{50,51}. The apparent activation energy for CO production was also
calculated as $125.9 \pm 7 \text{ kJ mol}^{-1}$, much higher than that of CH_3OH formation. Thus, the
mechanism of water gas shift reaction followed by CO hydrogenation should barely
contribute to the CH_3OH production by CO_2 hydrogenation reaction in our case, although it
could not be fully excluded at high CO_2 conversions." (Lines 151-161)

(2) On that note, at larger CO_2 (or CO) conversion, there will always be substantial water-
gas-shift or rWGS activity occurring. So, for example CO_2 might be converted to CO which
than is converted to methanol. It would be good to at least mention this dilemma at higher
conversions.

**Response:** We appreciate the reviewer's insightful comments very much. To address this
issue, we calculated and compared the apparent activation energies for CO_2 consumption,
CH_3OH production and CO production in the CO_2 hydrogenation reaction and for CO
consumption, CH_3OH production and CO_2 production in the CO hydrogenation reaction.
The results are shown below in Figure R2.

Figure R2. Arrhenius plots of (a) CO₂ consumption, CH₃OH production and CO production in
 the CO₂ hydrogenation reaction catalyzed by Zn20Zr80 catalyst within the kinetics-
 controlled range and (b) CO consumption, CH₃OH production and CO₂ production in the
 CO hydrogenation reaction catalyzed by Zn20Zr80 catalyst.

In the CO₂ hydrogenation reaction, the derived Arrhenius plots calculated from the CH₃OH
 formation and CO₂ reaction rates give similar apparent activation energy(E_a) of 81.9±1 and
 85.1±1 kJ mol⁻¹, respectively, while the apparent activation energy for CO production was
 also calculated as 125.9±7 kJ mol⁻¹, much higher than that of CH₃OH formation. Thus, the
 mechanism of water gas shift reaction followed by CO hydrogenation should barely
 contribute to the CH₃OH production by CO₂ hydrogenation reaction in our case, although it
 could not be fully excluded at high CO₂ conversions.

In the CO₂ hydrogenation reaction, the corresponding Arrhenius plots using CH₃OH
 formation and CO consumption rate give E_a of 54.5±6 and 54.3±6 kJ mol⁻¹, respectively. The
 apparent activation energy for CO₂ production was also calculated as 37.3±8 kJ mol⁻¹.
 Considering the low CO conversions and very low CO₂ selectivity in our case, the
 contribution of CO₂ production and its consequent hydrogenation to CO hydrogenation to
 methanol can be ignored.

Action

In the revised manuscript, we have added the above Figure R2 as Supplementary Figure 11
 and discussed this issue as the following:

"The CO₂ conversion of 9.1% at 573 K is close to the equilibrium value (around 9.3%) under
 the adopted reaction condition⁴⁹. We then measured the catalytic performance within the
 kinetics-controlled range (Supplementary Table 2). The derived Arrhenius plots (Fig. 2b and
 Supplementary Fig. 11) calculated from the CH₃OH formation and CO₂ reaction rates give
 similar apparent activation energy(E_a) of 81.9±1 and 85.1±1 kJ mol⁻¹, respectively, consistent
 with the literature results^{50,51}. The apparent activation energy for CO production was also
 calculated as 125.9±7 kJ mol⁻¹, much higher than that of CH₃OH formation. Thus, the
 mechanism of water gas shift reaction followed by CO hydrogenation should barely
 contribute to the CH₃OH production by CO₂ hydrogenation reaction in our case, although it
 could not be fully excluded at high CO₂ conversions." (Lines 151-161)

"The corresponding Arrhenius plots (Fig. 2d and supplementary Fig. 11) using CH₃OH
 formation and CO consumption rate give E_a of 54.5±6 and 54.3±6 kJ mol⁻¹, respectively. The
 apparent activation energy for CO₂ production was also calculated as 37.3±8 kJ mol⁻¹.

Considering the low CO conversions and very low CO₂ selectivity in our case, the reaction
pathway of CO₂ production and its consequent hydrogenation to methanol can be ignored
in the CO hydrogenation to methanol reaction." (Lines 164-169)

(3) The authors observe from their studies that "However, the activation energy of bri-
HCOO* hydrogenation reaction is much larger than the apparent activation energy of CO₂
hydrogenation to methanol reaction (78.9 kJ mol⁻¹), therefore, the high-barrier surface
reaction pathway is not likely to be responsible for the methanol synthesis from CO₂
hydrogenation catalyzed by Zn₂₀Zr₈₀ catalyst ... " and that "... 62.3 and 69.2 kJ mol⁻¹,
slightly lower than the apparent activation energy of CO₂ hydrogenation to methanol (78.9
109 kJ mol⁻¹). Thus, the low-barrier surface reaction pathway is predominantly responsible for
the methanol synthesis from CO₂ hydrogenation catalyzed by the Zn₂₀Zr₈₀ catalyst." I am
not sure that comparing those observed activation barriers with those in Figure 2 is totally
fair (see comment above), as this might be also largely due to differences in measurement
protocols. I would think, however, that the trends towards e.g. lower activation barriers for
CO hydrogenation would still hold.

**Response:** We appreciate the reviewer's insightful comments very much. As replied above,
during the revision, we re-evaluated the catalytic reaction of CO₂ hydrogenation reaction at
523, 533 and 548 K using 600 mg Zn₂₀Zr₈₀ catalyst diluted with 400 mg SiC to acquire
reliable data for reaction kinetic analysis. The corresponding Arrhenius plot gives an
activation barrier for CO₂ hydrogenation to methanol of 81.9±1 kJ mol⁻¹. Comparing the
activation energies of high-barrier and low-barrier surface reaction pathways respectively of
around 113.0 and 62.3-69.2 kJ mol⁻¹, it can be thus drawn that the low-barrier surface
reaction pathway is predominantly responsible for the methanol synthesis from CO₂
hydrogenation catalyzed by the Zn₂₀Zr₈₀ catalyst.

(4) Concerning the DFT calculations, there are no barriers reported for the H₂ dissociation,
e.g. over the Zn-O pair. For metal-oxides this might be an important rate determining step
and should be considered.

**Response:** We appreciate the reviewer's insightful comments very much. During the revision,
we have calculated the barrier for H₂ dissociation on the Zn-ZrO₂ surface. The results show
that H₂ preferentially dissociates heterolytically at the Zn_{Zr}-O pair of the Zn-ZrO₂ surfaces to
form H*-Zn_{Zr} and H*-O species with a barrier of 23.2 kJ mol⁻¹ (0.24 eV), which is very low.

**Action**

In the revised manuscript, the above result has been added as the following:

"H₂ preferentially dissociates heterolytically at the Zn_{Zr}-O pair of the Zn-ZrO₂ surfaces to
form H*-Zn_{Zr} and H*-O species with a low barrier of 23.2 kJ mol⁻¹." (Lines 351-353)

(5) The calculated gibbs free energy at 573 K is barely negative, so this must be close to
equilibrium (see comment above). Does the DFT method give the correct ΔH for CO and
CO₂ hydrogenation (e.g. for CO₂ + H₂ → methanol a ΔH of -49 kJ mol⁻¹)?

**Response:** We appreciate the reviewer's insightful comments very much. Table R2 below
lists the computed thermodynamic values of reactant and products at CO₂ hydrogenation

reaction condition in Fig. 5d (573 K, 3 MPa, H₂: CO₂ = 3) and the standard condition. Using
 these data, the $\Delta G = G(\text{CH}_3\text{OH}(\text{g})) + G(\text{H}_2\text{O}(\text{g})) - 3G(\text{H}_2(\text{g})) - G(\text{CO}_2(\text{g}))$ was calculated as -9.9 kJ
 142 mol⁻¹ at the reaction condition, close to the equilibrium, while the $\Delta H^\ominus = H^\ominus(\text{CH}_3\text{OH}(\text{g})) +$
 $H^\ominus(\text{H}_2\text{O}(\text{g})) - 3H^\ominus(\text{H}_2(\text{g})) - H^\ominus(\text{CO}_2(\text{g}))$ was calculated as -45.3 kJ mol⁻¹, close to the
 experimental value (-49.5 kJ mol⁻¹).

Table R2. Thermodynamic values of reactant and products. E_{DFT} and E_{ZPE} represent the
 computed potential energy and zero-point energy, respectively. (H-ΔTS)_{573K} and RTln(p/p[⊖])
 correspond to the thermodynamic corrections under reaction conditions. ΔH_{corr}[⊖] refers to
 the enthalpy correction under standard conditions. G and H[⊖] (H = E_{DFT} + E_{ZPE} + ΔH_{corr}[⊖]) denote
 the calculated Gibbs free energy under the reaction condition and the enthalpy under the
 standard condition, respectively. The enthalpy changes and entropies for CO, CO₂, H₂, and
 H₂O under reaction and standard conditions, as well as for CH₃OH under standard
 conditions, are obtained from the NIST-JANAF Thermochemical Tables. The enthalpy
 change and entropy of CH₃OH under the reaction condition are calculated from DFT. All
 energies are in the unit of kJ mol⁻¹.

	E _{DFT}	E _{ZPE}	Reaction condition			Standard condition	
			(573 K, 3 MPa, H ₂ : CO ₂ = 3)			ΔH _{corr} [⊖]	H [⊖]
			(H-ΔTS) _{573K}	RTln(p/p [⊖])	G		
CO ₂ (g)	-2217.23	29.91	-117.71	9.55	-2295.38	9.65	-2177.67
H ₂ O (g)	-1372.02	54.03	-101.31	-10.61	-1429.91	9.65	-1308.34
H ₂ (g)	-654.16	26.05	-69.47	14.47	-683.11	8.68	-619.43
CH ₃ OH (g)	-2914.81	131.22	-130.25	-10.61	-2924.46	10.61	-2772.98

**Action**

In the revised manuscript, we have added the above Table R2 as Supplementary Table 9
 and described as the following:

"The enthalpy changes and entropies for CO, CO₂, H₂, and H₂O under reaction and standard
 conditions, as well as for CH₃OH under the standard condition, are obtained from the NIST-
 JANAF Thermochemical Tables⁶⁰ and the enthalpy change and entropy of CH₃OH under the
 reaction condition are calculated from DFT⁶¹ (Supplementary Table 9)." (Lines 866-869)

(6) Experimentally there is a difference observed between bri-HCOO at 1590 and 1561 (the
 latter involving presumably oxygen from the lattice). Does DFT reproduce this difference of
 about 30 cm⁻¹ also when comparing bri-HCOO* on Zr-Zn and with a lattice O, at least
 qualitatively? I do understand though that the total numbers might differ by more than 30
 166 cm⁻¹ between experiment and theory.

**Response:** We appreciate the reviewer's insightful comments very much. The different bri-
 HCOO* species exhibit experimentally-observed C-O vibrational modes at 1590 and 1561
 169 cm⁻¹ and C-O vibrational modes at 1562 and 1549 cm⁻¹, as mentioned in the manuscript.
 The experimental-observed and calculated values show a similar trend, and the differences
 between the experimental-observed values and the calculated values are 29 and 13 cm⁻¹,

respectively. The texts discussing this issue in the manuscript is copied as the following:

**“The bri-HCOO*-a species and bri-HCOO*-b species exhibit the calculated C-O**
 **vibrational mode at 1562 and 1549 cm⁻¹, respectively, and consequently correspond to**
 **the experimentally-observed bri-HCOO*-1590 spectator and bri-HCOO*-1561 active**
 **species on Zr20Zn80.”** (Lines 597-599)

(7) Figure 4 shows temporal DRIFTS and then the time evolution of certain peaks (in b and f).
 However, I do not see the correlation of some of the peaks, e.g. the 2955 in b) is absent in
 a). Likewise, I do fail to see the 2896 and 2925 peaks shown in f) in figure e).

**Response:** We appreciate the reviewer’s insightful comments very much. We have re-
 plotted Figure 4 and related Supplementary Figures 16 and 28 with correctly-marked peak
 positions as the following:

Revised Figure 4

Revised Supplementary Figure 16

Revised Supplementary Figure 28

**Action**

In the revised manuscript, we have used the above re-plotted Figure 4 and Supplementary

Figures 16 and 28 to replace the corresponding original figures.

(8) The authors use the PBE functional for their DFT part, but seem not to include dispersion
interaction (e.g. through D3). Since this can usually be done with little extra computational
cost and may actually be important for larger adsorbates such as H₂COO, I wonder why this
has not been included and what the effect would be?

**Response:** We appreciate the reviewer's insightful comments very much. During the revision,
we carried out additional theoretical calculations of CO₂ hydrogenation to methanol
reaction using PBE functional with D3 including dispersion interaction. The results
summarized in Table R3 below show that the dispersion interaction does not affect the
computed reaction barriers much.

**Table R3.** The computed transition state energy (G_a , kJ mol⁻¹) of CO₂ hydrogenation to
methanol reaction using PBE functional with without and with D3 dispersion correction
(PBE-D3).

	w/o. d3	w. d3
G _a (TS1)	34.73	30.88
G _a (TS2)	30.88	28.95
G _a (TS3)	10.61	8.68
G _a (TS4)	76.22	80.08
G _a (TS5)	31.84	27.98
G _a (TS6)	86.84	82.98
G _a (TS1')	88.77	86.84
G _a (TS2')	150.52	151.48
G _a (TS3')	44.38	46.31
G _a (TS4')	16.40	13.51
G _a (TS5')	46.31	48.24
G _a (TS6')	2.89	2.89

**Action**

In the revised manuscript, we have added the above Table R3 as Supplementary Table 7
and described as the following:

“Additional theoretical calculations using PBE-D3⁵⁵ to include the dispersion interaction
(Supplementary Table 7) show that the dispersion interaction does not affect the computed
reaction energies much.” (Lines 397-399)

(9) In general, the SI has been done with great care. It would be good to add the different
contributions coming from the simulations in Table S4 though. E.g. instead of only giving ΔE
and ΔG , I would prefer all calculated quantities, e.g. ΔE , ZPE, ΔS . That way an interested
reader can easily comprehend how the numbers have been derived. Importantly, that would
also allow anyone to replot the ΔG diagram at other temperatures, if of interest.

**Response:** We appreciate the reviewer's insightful comments very much. As replied above,
we have listed the computed thermodynamic values of reactants and products in
Supplementary Table 9 of the revised manuscript and described the results with citations as
the following:

“Table R2. Thermodynamic values of reactant and products. E_{DFT} and E_{ZPE} represent the
computed potential energy and zero-point energy, respectively. $(H-\Delta TS)_{573\text{K}}$ and $RT\ln(p/p^\ominus)$
correspond to the thermodynamic corrections under reaction conditions. $\Delta H_{\text{corr}}^\ominus$ refers to
the enthalpy correction under standard conditions. G and H^\ominus ($H=E_{\text{DFT}}+E_{\text{ZPE}}+\Delta H_{\text{corr}}^\ominus$) denote
the calculated Gibbs free energy under the reaction condition and the enthalpy under the
standard condition, respectively. The enthalpy changes and entropies for CO, CO₂, H₂, and
H₂O under reaction and standard conditions, as well as for CH₃OH under standard
conditions, are obtained from the NIST-JANAF Thermochemical Tables. The enthalpy
change and entropy of CH₃OH under the reaction condition are calculated from DFT. All
energies are in the unit of kJ mol⁻¹

	E_{DFT}	E_{ZPE}	Reaction condition		Standard condition		
			(573 K, 3 MPa, H ₂ : CO ₂ = 3)		G	$\Delta H_{\text{corr}}^\ominus$	H^\ominus
			$(H-\Delta TS)_{573\text{K}}$	$RT\ln(p/p^\ominus)$			
CO ₂ (g)	-2217.23	29.91	-117.71	9.55	-2295.38	9.65	-2177.67
H ₂ O (g)	-1372.02	54.03	-101.31	-10.61	-1429.91	9.65	-1308.34
H ₂ (g)	-654.16	26.05	-69.47	14.47	-683.11	8.68	-619.43
CH ₃ OH (g)	-2914.81	131.22	-130.25	-10.61	-2924.46	10.61	-2772.98

“The enthalpy changes and entropies for CO, CO₂, H₂, and H₂O under reaction and standard
conditions, as well as for CH₃OH under the standard condition, are obtained from the NIST-
JANAF Thermochemical Tables⁶⁰ and the enthalpy change and entropy of CH₃OH under the
reaction condition are calculated from DFT⁶¹ (Supplementary Table 9).”

(10) I would recommend to only use one unit throughout the manuscript. For example, in
Figure 5, ΔG is given in eV, but (experimental and theoretical) Arrhenius plots are given in kJ
235 mol⁻¹.

**Response:** We appreciate the reviewer's kind suggestion very much. We have unified the
units to kJ mol⁻¹ in the revised manuscript.

**Author Reply to Reviewer #2's Comments**

This article reports on an in-depth study on the comparison between CO and CO₂
hydrogenation over Zn/ZrO₂ systems for the selective production of methanol. The authors
propose two different active systems for both reaction and also elucidate the reaction
mechanism by a combination of theoretical calculations and spectroscopy measurements.
The article is of clear relevance to the field and also important for the further progression to
obtain more selective, active and ultimately also stable catalyst materials for these important
conversions process.

**Response:** We appreciate the reviewer's effort in reviewing our manuscript, positive
recommendation and insightful comments very much. We have seriously considered the
comments and revised the manuscript accordingly. We hope that the revised manuscript
will be suitable for the publication.

(1) Unfortunately, I find the experimental basis on which the conclusions is based not that
strong as a deeper analysis of the experimental data, as reported in the main text and the SI,
is required. The spectroscopy data reported reveal that the "operando mode" only holds for
the IR spectroscopy data where a clear attempt is done by the authors to correlate the
different species observed with the selectivity and activity patterns changing with time on
stream. As a side comment, I have to note here that in the SI the fitting of some of the data
with a very steep background (Figures S26-S28) is not that accurate and certainly prone to
errors. Having stated this, it would be worthwhile to further corroborate on this
quantitative/semi-quantitative analysis.

**Response:** We appreciate the reviewer's insightful comments very much. In our study, we
mainly performed two types of DRIFTS measurements: one was temporal in situ DRIFTS
measurements of the working catalysts combined with online mass spectroscopy
measurements of gas-phase compositions, which, we believe, can be termed as "**Operando**
**characterizations**"; the other is temporal in situ DRIFTS measurements of the working
catalysts for elementary surface reaction kinetic analysis. We have carefully distinguished
both cases in the revised manuscript.

The peak fitting processes of temporal in situ DRIFTS spectra were carried out using the
"Peak Resolve" function of the "Omnic" software integrated with the infrared spectrometer.
Typically, a set of original temporal in situ DRIFTS spectra within a fixed wavenumber range
for a specific vibrational mode was loaded and automatically processed by selecting a
"linear" baseline correction, a "Gaussian/Lorentzian" peak shape, a position constrained
within a reasonable range and a fixed full width at half maximum (FWHM) of the peak of the
same species to acquire the fitted spectra. Such a data analysis process integrated with the
infrared spectrometer should give reliable results. Moreover, the backgrounds of the same
set of temporal in situ DRIFTS spectra are similar. Thus, we believe that our peak-fitting
results of the same set of temporal in situ DRIFTS spectra are accurate enough for the
experimental establishments of reliable elementary surface reaction kinetics, which are
supported by the accompanying theoretical simulations.

During the revision, we also re-analyzed the temporal in situ DRIFTS spectra of Zn₂₀Zr₈₀

(pretreated in 3 MPa CO₂ at 303 K for 60 min, then purged in Ar) exposed to 3 MPa H₂
at 443, 453 and 463 K, the spectra commented by the reviewer, by firstly subtracting the linear
background and then doing the peak-fitting processes with the “Omnici” software. The
peak-fitted temporal in situ DRIFTS spectra are shown below in Figure R3, whose results
give the evolutions of different surface species as a function of time in the low-temperature
CO₂ hydrogenation to methanol reaction pathway (Figure R4 below). Kinetic analysis similar
to that adopted in Figure 4 f-h gives the barrier of CH₃O* formation between 66.1±9 and
70.2±2 kJ mol⁻¹, similar to the results of 62.2±9~69.2±1 kJ mol⁻¹ acquired using the
automatic peak-fitting process. This also demonstrate the reliability of the automatic peak-
fitting process adopted in the manuscript.

Figure R3. Peak-fitted time-resolved in situ DRIFTS spectra of Zn₂₀Zr₈₀ exposed to 3 MPa
 H₂ (pretreated in 3 MPa CO₂ at 303 K for 60 min, then purged in Ar) at (a1 to a4) 443 K, (b1
 to b4) 453 K and (c1 to c4) 463 K.

Figure R4. The corresponding normalized integrated peak area of various species in Figure
 R5 as a function of time on Zn₂₀Zr₈₀ catalyst at (a) 443 K, (b) 453 K and (c) 463 K.

Figure R5. (a) the average rate of CH₃O* formation reaction on Zn₂₀Zr₈₀ at 0.8 to 2.4 min
 derived from Figure R6. (b) Arrhenius plot of CH₃O* formation reaction using the CH₃O*
 formation rate at 0.8 min and the average rate of 0.8-2.4 min.

Action

In the revised manuscript, we have re-written the caption of Figure 6 and described the
 peak-fitting processes of DRIFTS spectra as the following:

"Fig. 6. Operando characterizations and in situ restructuring in CO hydrogenation
reaction." (Lines 424-425)

"The peak fitting processes of temporal in situ DRIFTS spectra were carried out using the
"Peak Resolve" function of the "Omnic" software integrated with the infrared spectrometer.
Typically, a set of original temporal in situ DRIFTS spectra within a fixed wavenumber range
for a specific vibrational mode was loaded and automatically processed by selecting a
"linear" baseline correction, a "Gaussian/Lorentzian" peak shape, a position constrained
within a reasonable range and a fixed full width at half maximum (FWHM) of the peak of the
same species to acquire the fitted spectra." (Lines 781-787)

(2) I have to disagree with the authors on the notion that Figure 6 would suggest that an
"operando/in-situ mode" is achieved; as stated in the figure caption. I also would not call
the EPR data even quasi in-situ. The samples are just "trapped" from the reaction
environment and cooled down be measured at the low temperature required to measure
the EPR spectra (e.g., 140 K). The entire EPR data have to be investigated in much more
depth. Here, the authors have first to show the entire range from 0 to 6000 G (or something
like this) and focus on the different signals observed. I am curious to see if there are other
regions of interests and what the impurities are in the support material.

**Response:** We appreciate the reviewer's insightful comments very much. As described in
the experimental section, a high-pressure stainless-steel reaction tube mounted with two
valves that could be closed to seal the powder sample from ambient atmosphere was used
for ESR measurements. After CO₂ or CO hydrogenation reactions at desirable conditions
and cooling to 303 K in the reaction atmospheres, the reaction tube was sealed, removed
from the reactor, transferred to an Ar-filled glovebox (Mikrouna Co. Ltd., China, O₂
concentrations below 0.1 ppm), where it was opened and 100 mg of the catalyst sample
were transferred and sealed into an electron spin resonance (ESR) sample tube for the ESR
measurements at 140 K on a JEOL JESFA200 ESR spectrometer (9.087GHz) at a microwave
power of 0.998 mW, a modulation frequency of 100 kHz, and a modulation amplitude of
0.35 mT. Thus, the ESR measurements were carried out for the catalysts treated in reaction
conditions without exposure to air.

All the ESR spectra were measured within the 2845 to 3645 G range and only the signals of
O₂⁻, F⁺ centers, Zr³⁺ and interstitial Zn_i⁺ defects were observed (Figure R6 below). Following
the reviewer's suggestion, we measured an ESR spectrum of Zn₂₀Zr₈₀ calcined in Ar at 773
K without exposure to air within the 0 to 6000 G range (Figure R7a below), which shows no
obvious signals; however, an ESR spectrum of the same sample within the 3250 to 3500 G
range (Figure R7b below) gives the signals of O₂⁻, F⁺ centers, Zr³⁺ and interstitial Zn_i⁺ defects,
similar to the ESR spectrum in the manuscript. We were told that the adopted JEOL JES-
FA200 ESR spectrometer outputs a fixed 8192 data points no matter how wide the test
ranges are, thus a wide range ESR measurement of the 0 to 6000 G range is not good at
characterizing the paramagnetic species with narrow ESR signals such as the O₂⁻, F⁺ centers,
Zr³⁺ and interstitial Zn_i⁺ defects in our case, but good at probing the transition metal
paramagnetic ions with *d*-orbitals which have strong coupling effects. Thus, demonstrated
by the wide-range ESR spectrum (Figure R7a), our Zn₂₀Zr₈₀ sample does not have

transition metal impurities.

Figure R6. ESR spectra in the range of 2845 to 3645 G of Zn20Zr80 (a) calcined in Ar at 773
K(Zn20Zr80-Ar), and subsequently treated under 3 MPa CO+H₂ (H₂:CO=2) (Zn20Zr80-
CO+H₂) and under 3 MPa CO₂+H₂ (H₂: CO₂=3) (Zn20Zr80-CO+H₂) without exposure to air.
(b) calcined in Ar at 773 K(Zn20Zr80-Ar), and subsequently treated under 3 MPa CO+H₂
(H₂:CO=2) (Zn20Zr80-CO+H₂) without exposure to air, and then exposed to air (Zn20Zr80-
CO+H₂-Air). (c) calcined in Ar at 773 K, and subsequently treated under 3 MPa CO+H₂
(H₂:CO=2) at different temperatures without exposure to air. All the spectra were measured
at 140 K.

Figure R7. ESR spectra of Zn20Zr80 calcined in Ar at 773 K without exposure to air
measured within (a) the 0 to 6000 G range and (b) the 3250 to 3500 G range at 130 K.

Action

In the revised manuscript, we have clarified that our ESR measurements were carried out
without exposure to air and removed the “quasi in situ” term as the following:

“Electron paramagnetic resonance (ESR) spectra of Zn20Zr80 calcined in Ar at 773 K without
exposure to air (Fig. 1d and Supplementary Fig. 2)” (Lines 90-91)

“ESR spectra of Zn20Zr80 calcined in Ar at 773 K, and subsequently treated under 3 MPa
CO+H₂ (H₂:CO=2) or 3 MPa CO₂+H₂ (H₂: CO₂=3) at 573 K without exposure to air.” (Line 104)

“...without exposure to air...” (Line 171)

“...without exposure to air...” (Line 430)

“...without exposure to air...” (Line 436)

“...without exposure to air...” (Line 465)

Meanwhile, we have also added the above Figure R7a as Supplementary Figure 2 and
described this issue as the following:

“...and Supplementary Figure 2...” (Line 91)

“The adopted JEOL JES-FA200 ESR spectrometer outputs a fixed 8192 data points no matter
how wide the test ranges are, thus a wide range ESR measurement of the 0 to 6000 G range
is not good at characterizing the paramagnetic species with narrow ESR signals such as the
O_2^- , F^+ centers, Zr^{3+} and interstitial Zn_i^+ defects in our case (Figure 1d), but good at probing
the transition metal paramagnetic ions with d -orbitals which have strong coupling effects.
Thus, the wide-range ESR spectrum shown here suggests that our Zn₂₀Zr₈₀ sample does
not have transition metal impurities.” (Supplementary information Lines 44-50)

(3) Regarding the identification of the different signals, it is important to plot their intensity
as a function of the measurement temperature to see if they have Curie-Weiss law
characteristics and are pragmatically isolated sites.

**Response:** We appreciate the reviewer's insightful comments very much. Following the
reviewer's suggestion, we studied the temperature dependence of the inverse EPR
susceptibility χ_{EPR} of the ESR signals of Zn₂₀Zr₈₀ calcined in Ar at 773 K without exposure to
air at the constant magnetic field modulation (3.500 G) and radio frequency power (2.000
389 mW). As shown in Figure R8 below, the ESR signals continuously decrease with the
390 measuring temperature of ESR spectra increasing. The O_2^- and Zn_i^+ ESR signals could not be
detected when the measurement temperatures rose to 150 and 180 K, respectively. The
plots of χ_{EPR} of observed F^+ centers, Zr^{3+} , and Zn_i^+ ESR signals against the measurement
temperatures according to the method previously reported by N. Guskos and co-workers
(Guskos, N. *et al.* Cooperative magnetic interactions in three mononuclear copper Schiff
base complexes. *Mater. Chem. Phys.* **83**, 114-119 (2004)) follow the Curie-Weiss law,
confirming that the observed ESR signals arise from the pragmatically isolated sites.
However, the plot of χ_{EPR} of observed O_2^- ESR signals against the measurement temperatures
could not be made because it had only two data.

Figure R8. (a) ESR spectra of Zn₂₀Zr₈₀ calcined in Ar at 773 K without exposure to air
 measured at different temperature at the constant magnetic field modulation (3.500 G) and
 radio frequency power (2.000 mW). Temperature dependence of the inverse EPR
 susceptibility χ_{EPR}^{-1} of the three observed ESR signals of (b) F⁺ centers, (c) Zn²⁺ and (d) Zr³⁺.

Action

In the revised manuscript, we have added the above Figure R8 as Supplementary Figure 3
 and discussed this issue as the following:

"The inverse EPR susceptibility χ_{EPR} of the observed ESR signals measured at different
 temperatures (Supplementary Fig. 3) were found to follow the Curie-Weiss law^{47,48},
 confirming that these ESR signals arise from the pragmatically isolated sites." (Lines 92-94)

(4) Also, some EPR simulations would be here advantageous. The authors argue that the
 addition of O₂ leads to a disappearance of the signal but that may not wonder as O₂ is a
 paramagnetic molecule itself. Furthermore, the authors have to further analyze in depth -
 via a systematic analysis of the different reaction and measurement parameters - the EPR
 spectra what can be learned from this. These data as they are not in-situ or operando
 makes that only static information can be obtained. I am also a bit surprised that the plot in
 Figure 6e is with an x-axis which is relative (not absolute) and only ranging between 1 and 2,
 while the two y-axis have a 0 as the lowest value. This plot is not understandable to me and
 the question is how accurate this is and how it is exactly obtained (knowing the difficulties
 associated with the EPR measurements and the different species observed).

**Response:** We appreciate the reviewer's insightful comments very much. As described in
 the manuscript, the ESR spectra of treated Zn₂₀Zr₈₀ catalysts without exposure to air and
 then exposed to air were measured, and we consider that the ESR signals quenched after
 exposure to air locate on the catalyst surface but those not locate in the bulk. ESR spectrum
 of Zn₂₀Zr₈₀ calcined in Ar at 773 K without exposure to air (Fig. 1d and Supplementary Fig.
 2) only show signals of O₂⁻, F⁺ centers, Zr³⁺ and interstitial Zn²⁺ defects, and these ESR

features barely change after an exposure to air (Supplementary Fig. 4), suggesting that the
 F^+ centers, Zr^{3+} and interstitial Zn_i^+ defects locate in the bulk. ESR spectrum of Zn20Zr80
 subjected to the CO hydrogenation reaction at 573 K without exposure to air exhibits
 substantially increased signals of O_2^- and F^+ centers but barely changed signals of Zr^{3+} and
 Zn_i^+ defects (Fig. 1d). Following a subsequent exposure to air, the signals of O_2^- and F^+
 centers are quenched while those of Zr^{3+} and Zn_i^+ defects do not change (Supplementary
 Fig. 39), suggesting that the increased O_2^- and F^+ centers on Zn20Zr80 subjected to the CO
 hydrogenation reaction at 573 K should probably locate on the Zn20Zr80 surface. We
 believe that such an analysis is a reasonable approach. The quench of O_2^- and F^+ centers on
 Zn20Zr80 subjected to the CO hydrogenation reaction at 573 K upon exposure to air might
 be due to CO_2 and/or water adsorption rather than O_2 adsorption.

We are sorry that we did not define the x axis in Figure 6e in the original submission, which
 was calculated by $(I(O_2^-+F^+ \text{ ESR signals})_{CO+H_2 \text{ reaction}}/I(O_2^-+F^+ \text{ ESR signals})_{Ar})$ to represent the
 normalized amount of created O_2^- and F^+ centers on Zn20Zr80 subjected to CO
 hydrogenation reaction at different temperatures. The intensity (I) of O_2^- and F^+ centers was
 acquired by integrating their peak areas in the absorption curves obtained by an integral
 treatment of the ESR spectrum (*Electrochim. Acta* **53**, 4580-4590 (2008); *J. Macromol. Sci.,*
 *Part A* **45**, 195-198 (2008)). After considering the reviewer's comments, we re-plot Figure 6e
 more appropriately using the $(I(O_2^-+F^+ \text{ ESR signals})_{CO+H_2 \text{ reaction}} - I(O_2^-+F^+ \text{ ESR signals})_{Ar})$, the
 amount of created O_2^- and F^+ centers on Zn20Zr80 subjected to CO hydrogenation reaction
 at different temperatures, as the x axis (Figure R9 below). The results also show that the
 amount of created O_2^- and F^+ centers signals on the Zn20Zr80 catalyst surface subjected to
 CO hydrogenation at different temperatures is proportional to the corresponding intensity of
 vibrational feature of $bri-HCOO^*-1561$ species and the corresponding CH_3OH formate
 rate.

Figure R9. Intensity of vibrational feature of $bri-HCOO^*-1561$ species on Zn20Zr80 and
 CH_3OH formate rate of Zn20Zr80-catalyzed CO hydrogenation reaction as a function of
 corresponding amount of created O_2^- and F^+ centers signals on Zn20Zr80 surface.

**Action**

In the revised manuscript, we have re-written the texts describing and discussing the ESR

spectra changes after Zn20Zr80 subjected to the CO hydrogenation reaction at 573 K was
exposed to air as the following:

"...whereas those of Zr^{3+} and Zn^{2+} defects do not change (Supplementary Fig. 39), suggesting
that the increased O_2^- and F^+ centers on Zn20Zr80 subjected to the CO hydrogenation
reaction at 573 K should probably locate on the Zn20Zr80 surface." (Lines 439-441)

Meanwhile, we have replaced the original Figure 6e with the above Figure R9 and described
the results as the following:

"The amount of paramagnetic centers in a sample is proportional to the area under its
absorption curve that can be obtained by an integral treatment of the ESR spectrum^{56,57},
following which the intensities of O_2^- and F^+ centers signals on Zn20Zr80 calcined in Ar at
773 K ($I(O_2^- + F^+ \text{ ESR signals})_{Ar}$) and subsequently treated under 3 MPa $CO+H_2$ at different
temperatures ($I(O_2^- + F^+ \text{ ESR signals})_{CO+H_2 \text{ reaction}}$) were acquired. The intensity of vibrational
feature of $bri-HCOO^*-1561$ species on Zn20Zr80 treated under 3 MPa $CO+H_2$ at different
temperatures and the CH_3OH formate rate of Zn20Zr80-catalyzed CO hydrogenation
reaction at different temperatures were found proportional to the corresponding amount of
created O_2^- and F^+ centers on Zn20Zr80 surface, represented by $I(O_2^- + F^+ \text{ ESR signals})_{CO+H_2}$
$_{reaction} - I(O_2^- + F^+ \text{ ESR signals})_{Ar}$ (Fig. 6e)." (Lines 467-476)

(5) The comment on the operando/in-situ mode of operation for EPR also holds for the
measured/reported XAS (EXAFS and XANES) as well as the HRTEM as they also are taken
under "fixed" conditions, far from the real reaction conditions. Hence, the IR spectra are the
only one which provides something about the reaction mechanism, but not that much can
be stated about the active sites in the two set of reactions/systems.

**Response:** We appreciate the reviewer's insightful comments very much. In our work, we
firstly used temporal in situ DRIFTS to acquire the barriers of different surface reaction
pathways of Zn20Zr80-catalyzed CO_2 or CO hydrogenation to methanol reactions, which,
comparing the corresponding apparent activation energies, reliably identifies the dominant
surface reaction pathway and surface formate intermediate contributing to the apparent
catalytic activity. Then these results, combined with structural information provided by ESR
results of Zn20Zr80 treated in reaction conditions without exposure to air, HRTEM and XAS,
were used for a comprehensive theoretical simulation to propose the active sites involved in
the dominant surface reaction pathways. The core of our approach is the combination of
experimental elementary surface reaction kinetics acquired by temporal in situ DRIFTS and
comprehensive theoretical simulation, in which experimental elementary surface reaction
kinetics acts to judge the right model among various theoretical simulation results. The
structural information characterized with ESR, HRTEM and XAS mainly tell us the minimum
number of structural models needed to be considered in the theoretical simulations. Thus,
we did not struggle to do in situ XAS measurements of working Zn20Zr80 catalysts and
HRTEM characterizations of Zn20Zr80 catalysts treated in different conditions. Meanwhile,
our Zn20Zr80 catalyst turns out to be is a bulk oxide catalyst with non-uniformly-distributed
Zn species, and the roles of in situ XAS measurement and HRTEM characterizations are
limited. However, we are working on the stability of $Zn_{1/m}-ZrO_2$ single atom catalyst briefly
mentioned in the manuscript in CO_2 or CO hydrogenation reactions, in which in situ XAS

measurement and HRTEM characterizations play a key role. But this will be another story
hopefully to be reported in the near future.

(6) The comments above bring me to the more general question that one may question
why CO mostly formed from CO₂ requires such a different reaction site in this process
relative to each other and that one expect a first reaction between CO₂ towards CO and
then further from CO to methanol. I can expect that there two routes and hence two active
phases, but the question is then if they are both not interrelated and why would site isolated
be needed for one route and a cluster-type active site for the other route. This is not clear
when reading the paper.

**Response:** We appreciate the reviewer's insightful comments very much. There are two key
differences between CO₂ hydrogenation to methanol and CO hydrogenation to methanol
reactions on our Zn₂₀Zr₈₀ catalyst with co-existing Zn₁ single atom exclusively with the Zn-
O-Zr local structure and Zn_n clusters with both the Zn-O-Zr and Zn-O-Zn local structures.
One is that the catalyst surface is not reduced during CO₂ hydrogenation to methanol
reaction whereas the Zn_n clusters on the catalyst surface are reduced to Zn_{n,ov} clusters during
CO hydrogenation to methanol reaction at which CO hydrogenation to methanol reaction
proceeds with the lowest barrier. The other is that the bri-HCOO* species commonly
observed on Zn₂₀Zr₈₀ during CO₂ or CO hydrogenation to methanol reaction probably are
with different structures although they exhibit similar vibrational features. On one hand,
both O atoms of bri-HCOO* species come from CO₂ for CO₂ adsorption while one O atom
comes from CO and the other comes from Zn₂₀Zr₈₀ surface for CO adsorption; on the
other hand, the local structure of surface site on Zn₂₀Zr₈₀ binding the bri-HCOO* species
is different. Thus, the formation and surface reactivity of bri-HCOO* species vary with their
structures. These key differences lead to different active sites and reaction mechanisms of
Zn₂₀Zr₈₀ catalyst in catalyzing CO₂ hydrogenation to methanol and CO hydrogenation to
methanol reactions.

Action

In the revised manuscript, we have clarified this issue as the following:

"It is noteworthy that the CO hydrogenation to methanol reaction pathway at the in situ
formed Zn_{n,ov} active site does not exist in the Zn₂₀Zr₈₀-catalyzed CO₂ hydrogenation
reaction due to the absence of Zn_{n,ov} active site. This exemplifies the high sensitivity of
surface structure of working catalyst to the reaction atmosphere and consequently the
active site structure. It is also noteworthy that the bri-HCOO* species observed on
Zn₂₀Zr₈₀ during CO₂ or CO hydrogenation to methanol reaction are with different
structures and consequently formation ability and surface reactivity although they exhibit
similar vibrational features. On one hand, both O atoms of bri-HCOO* species come from
CO₂ for CO₂ adsorption while one O atom comes from CO and the other comes from
Zn₂₀Zr₈₀ surface for CO adsorption; on the other hand, the local structure of surface site
on Zn₂₀Zr₈₀ binding the bri-HCOO* species is different. The bri-HCOO* species is
commonly observed as a surface intermediate in the vibrational spectra characterizing oxide
catalysts for CO₂ or CO hydrogenation to methanol reaction, but its structures and surface
reactivity need to be carefully identified in order to determine the role." (Lines 607-620)

Author Reply to Reviewers' Comments

Reviewer #1 (Remarks to the Author):

The authors improved the manuscript considerably, addressing all my concerns. I therefore recommend to accept the manuscript for publication.

Reviewer #2 (Remarks to the Author):

I have now read the revised version of the paper, as well as the rebuttal letter. The authors have done a real effort to address my initial comments and also have made sure that a more quantitative approach is taken into the analysis of the spectroscopic data and also have nuanced where needed the aspect of the operando/in-situ/fixed reaction environment conditions thereby showing the strength and weaknesses of their analytical approaches taken. As far as I can judge, I do believe the article is now of sufficient quality and novelty to be published in Nature Communications and hence I propose it to be accepted as such.

Response: We appreciate the reviewers' effort in reviewing our revised manuscript and positive recommendation very much.